# Coupling slope-area analysis, integral approach and statistic tests to steady state bedrock river profile analysis

Yizhou Wang, Huiping Zhang, Dewen Zheng, Jingxing Yu, Jianzhang Pang, and Yan Ma

State Key Laboratory of Earthquake Dynamics, Institute of Geology, China Earthquake Administration, Beijing 100029, China

*Correspondence to*: Yizhou Wang (wangyizhou.-123@163.com)

**Abstract.** Slope-area analysis and the integral approach both have been widely used in stream profile analysis. The former is better at identifying changes in concavity indices but produces stream power parameters with high uncertainties relative to the integral approach. The latter is much better for calculating channel steepness. Limited work has been done to couple the advantages of the two methods and to remedy such drawbacks. Here we show the merit of the log-transformed slope-area plot to determine changes in concavities and then to identify colluvial, bedrock and alluvial channels along river profiles. Via the integral approach, we obtain bedrock channel concavity and steepness with high precision. In addition, we run bi-variant linear regression statistic tests for the two methods to examine and eliminate serially correlated residuals because they may bias both the estimated value and precision of stream power parameters. We finally suggest that, the coupled process, integrating the advantages of both slope-area analysis and the integral approach, can be a more robust and capable method for bedrock river profile analysis.

## 1 Introduction

In an evolving landscape, information about tectonics, climatic change, and lithology can be recorded by the bedrock river profiles (Fox et al., 2014, 2015; Goren et al., 2014; Harkins et al., 2007; Royden and Perron, 2013; Snyder et al., 2000). How to retrieve such details has long been a focus in both geologic and geomorphologic researches (Flint, 1974; Wobus et al., 2006; Rudge et al., 2015). Most of these studies are based on a well known power-law relationship between local channel gradient and drainage area (Flint, 1974; Hack, 1973; Howard and Kerby, 1983):

$$\frac{dz}{dx} = k_s A^{-\theta} \tag{1}$$

$$k_s = (U/K)^{1/n} \tag{2}$$

$$\theta = m/n \tag{3}$$

where $z$ is elevation, $x$ is horizontal upstream distance, $U$ is bedrock uplift rate, $K$ is an erodibility coefficient, $A$ is drainage area, and $m$ and $n$ are constants. Parameters $\theta$ and $k_s$ are referred to as concavity and steepness indices, respectively. The power-law scaling holds only for drainage areas above a critical threshold, $A_{cr}$, which is the transition from divergent to

convergent topography or from debris-flow to fluvial processes (Montgomery and Foufoula-Georgiou, 1993; Tarboton et al., 1989; Wobus et al., 2006). A growing number of studies have quantitatively related steepness to rock uplift (Hu et al., 2010; Kirby and Whipple, 2012; Kirby et al., 2003, 2007; Tarboton et al., 1989). Assuming a steady state river profile under constant rock uplift rates and erodibility in time and space, two forms of solutions to Eq. (1) are derived:

$$\log(\frac{U}{K})^{1/n} + \left(-\frac{m}{n}\right)\log A = \log\left(\frac{dz}{dx}\right) \tag{4}$$

$$z = z_b + (\frac{U}{KA_0^m})^{1/n}\chi \tag{5}$$

$$\chi = \int_0^x (\frac{A_0}{A(x')})^{m/n}\,dx' \tag{6}$$

where $z_b$ is the channel elevation at $x=0$ (river outlet). This is a boundary condition to Eq. (1). $A_0$ is an area-scale factor.

The slope-area analysis, as shown in Eq. (4), yields concavity and steepness indices by a linear fit to the log-transformed slope-area plot. Concavity changes with different channel substrate properties, which can be reflected and extracted from the slope-area data directly. Then, one can discriminate channel properties according to variable concavity indices. For example, available studies indicate that the colluvial, bedrock and alluvial channels can be directly identified from the log-transformed slope-area plot (Kirby et al., 2007; Snyder et al., 2000, 2003; Wobus et al., 2006). However, estimates of slope obtained by differentiating and resampling noisy elevation data are even noisier (Perron and Royden, 2013). Differentiation leads to considerable scatter in slope-area plots, making it challenging to identify a power-law trend with adequate certainty (Perron and Royden, 2013). In addition, the derived channel steepness suffers from high uncertainty due to error propagation (Perron and Royden, 2013; see Sect. 3 for details).

The integral approach, based on an integration of Eq. (1), was proposed by Royden et al (2000) to alleviate such problems by avoiding calculating channel slope. As shown in Eqs. (5) and (6), the transformed variable χ can be determined directly from drainage area data by simple numerical integration. Based on a proper concavity, the steady state river profile can be converted into a straight line. Slope of the line is steepness (we assume $A_0$=1 $m^2$ throughout the paper). As the best fit value of $\theta$ is not known a priori, we can compute χ-$z$ plots for a range of $\theta$ values and test for linearity (Perron and Royden, 2013). Thus, the integral method provides an independent constraint on both $\theta$ and $k_s$ (Perron and Royden, 2013). Nevertheless, the χ transformation contains an assumption of a single concavity, which is distinctly different from slope-area analysis. In fact, concavity can change. In places where there is spatially varying concavities (because channels may go from bedrock to alluvial), the integral approach may show a break in the χ-$z$ plot. Methods of separating areas of different concavities from a χ-$z$ plot have not been suggested. Despite a very noisy method compared to the integral approach, slope-area analysis is a more direct measure of concavity, because unlike the integral method one does not have to set an $m/n$ ratio, but rather measures this ratio directly from topographic data. In addition, the uncertainty in $k_s$ will be underestimated using the integral method, because the transformed profile (χ-$z$ plot) is a continuous curve, and therefore the residuals of the linear fit are serially correlated (Perron and Royden, 2013).

Based on the analysis above, coupling the advantages of the two methods can make up for their individual drawbacks and provide a better way to constrain stream power parameters. We also run bivariate linear regression statistic tests for the two methods to evaluate if the residuals of linear fit are homoscedastic and serially correlated. In this paper, we take streams, located in the Mendocino Triple Junction (MTJ) region of northern California (Fig.1), for example, to illustrate the process.

## 2 Methods

### 2.1 Coupling slope-area analysis and the integral approach

A natural river usually consists of different channel substrates, for example, colluvial, bedrock and alluvial channels. In spite of their complex formation processes, we can identify them from a log-transformed slope-area plot (Fig. 1) (Snyder et al., 2000). The colluvial channel, characterized by steep channel slope ($>20^{o}$) and limited drainage area ($<A_{cr}$) (Wobus et al., 2006), is frequently debris flow dominated and therefore will not display the typical fluvial scaling in Eq. (1) (Stock and Dietrich, 2003). Both bedrock (detachment-limited) and alluvial (transport-limited) channels show descending gradient with increasing drainage areas, which often exhibit a power-law scaling (Whipple and Tucker, 1999; Willgoose, 1994). However, the alluvial channel is often characterized by much gentler gradient and a higher concavity (Kirby et al., 2007; Snyder et al., 2000; Whipple and Tucker, 2002), which can be distinguished in the log-transformed plot (Fig. 1).

Via the integral approach (Perron and Royden, 2013), we derive concavities of bedrock channels. Based on a reference concavity index (Hu et al., 2010; Kirby et al., 2003, 2007; Perron and Royden et al., 2013; Snyder et al., 2000;Wobus et al., 2006), the $\chi$-$z$ analysis of the channel can be derived from the bedrock section that is suggested by the varying concavities in slope-area space.

### 2.2 Statistic tests

The coupled process does provide a better way to perform stream profile analysis. Indeed, both the slope-area analysis and integral approach are bi-variant linear regression methods. Statistically, some tests must be done to meet two critical conditions, i.e. the residuals are independent and homoscedastic (Cantrell, 2008; Kirchner, 2001). Perron and Royden (2013) noticed that the precision in steepness derived from the integral approach would be overestimated due to auto-correlated residuals. Mudd et al (2014) proposed a statistical framework to quantify spatial variation in channel gradients and calculated Durbin-Watson statistics in their code (https://csdms.colorado.edu/wiki/Model:Chi_analysis_tools). In this contribution, we find that auto-correlation of residuals may bias the regression coefficient, channel steepness (see details in Sect. 3). Therefore, not only theoretically but in practice, statistical tests are necessary for both the two methods. Here, we combine the Durbin-Watson test (Durbin and Watson, 1950) and Spearman rank correlation coefficient test (Choi, 1977; Fieller et al., 1957; York, 1968) to examine if the residuals are independent and homoscedastic. These tests are performed on the sections identified as bedrock using the slope-area analysis.

**2.2.1 Durbin-Watson test**

We took the integral approach for an example and rewrote Eq. (5) into another form:

$$z_i = z_b + k_s\chi_i + e_i \ (i = 1, 2 \cdots p) \tag{7}$$

In the formula, $p$ is the number of elevation data points, and $e$ represents residuals. We determined the Durbin-Watson (DW) statistics in the following steps:

    1) We first calculated the self-correlation coefficient of residuals via Eq. (8):

$$r = \frac{\sum_{i=2}^{p} e_i e_{i-1}}{\sqrt{\sum_{i=2}^{p} e_i^2}\sqrt{\sum_{i=2}^{p} e_{i-1}^2}}, \tag{8}$$

    2) Then, the DW statistic was derived as: DW=2×(1-$r$). Since -1≤r≤1, DW falls in the range of 0-4.

    3) We then examined if the residuals were auto-correlated according to Table 1.

To eliminate the self-correlation, new variables were generated as Eq. (9):

$$z_i' = z_i - rz_{i-1}, \ \chi_i' = \chi_i - r\chi_{i-1}(i = 1,2 \cdots p), \tag{9}$$

Slope of a linear fit to revised relative elevation, $z'$, and $\chi'$data are channel steepness.

**2.2.2 Spearman rank correlation coefficient test**

To evaluate if the variance of residuals is a constant, we utilized Spearman rank correlation coefficient test (Choi, 1977; York, 1968):

    1) Via a linear regression of χ-$z$ plots, we derived the absolute values of residuals |$e$|;

    2) We sorted the χ values in descending order and recorded the ranks $d_{i-1}$. Then the χ values were sorted again according to |$e$| and the new ranks were recorded as $d_{i-2}$;

    3) The Spearman rank correlation coefficient, $rs$, and the t-statistics, $t$, were calculated via Eqs. (10) and (11):

$$rs = 1 - \frac{6}{p(p^2-1)}\sum_{i=1}^{p}(d_{i-1} - d_{i-2})^2 \tag{10}$$

$$t = \frac{\sqrt{p-2}}{\sqrt{1-rs^2}}rs \ \ (i = 1,2 \cdots p) \tag{11}$$

    4) When the $t$ value is lower than a threshold, $t_{\alpha/2}$ ($p$-2), the variance of residuals is a constant. In our example, with $p > 30$ and significance level α = 0.05, the threshold value is larger than 2.58.

# 3 Case study: Mendocino Triple Junction (MTJ) region

Based on 1 arc-second SRTM DEM (digital elevation model), we extracted 15 streams in Mendocino Triple Junction (MTJ) region (Fig. 2). Here we first took streams Cooskie and Juan, for example, to illustrate the advantages and disadvantages of slope-area analysis and the integral approach, as well as to explain the reason of coupling the two methods.

Channel concavity and steepness indices can be derived from either slope-area analysis or the integral approach. For the same river profile, both methods should yield identical results (Scherler et al., 2014). We divided the profile of Cooskie stream into colluvial and bedrock channels from the log-transformed slope-area plot by eye (Fig. 3a). The area of process transition along a river profile can be determined by some rigorous methods. For example, Mudd et al (2014) used a segmentation algorithm and Clubb et al (2014) used a two-segment method for 1st order channels to find the area of process transition. Nevertheless, Fig. 3a shows a very simple log-transformed slope-area plot, from which the colluvial (nearly constant log(slope) ~ -1) and fluvial (decreasing channel gradient) sections can be discriminated just by eye. The elevation and area of the dividing point are ~500 m (Fig. 3b) and 0.1 km$^2$ (critical area, $A_{cr}$). The concavity of bedrock channel is ~0.47±0.05. We also computed the correlation coefficients between bedrock channel elevation and χ values based on a range of $θ$ values. The best linear fit corresponds to $θ$=0.45. Both of them are similar to the result (0.43±0.12) of Snyder et al (2000), but slightly higher than the result (0.36) of Perron and Royden (2013), which may be attributed to the difference in DEM resolution or choosing different critical areas.

Although the concavities derived from the two methods are in agreement, uncertainties (dividing the estimated value by error) in channel steepness differ a lot. The uncertainty from slope-area analysis is ~40% ($k_s$=79.16±29.35) (Fig. 3a), but the integral approach gives only ~0.5% ($k_s$=62.81±0.39) (Fig. 3d). In addition to smoothing and re-sampling of elevation data, we attribute such large uncertainty to error propagation. The natural logarithmic value of steepness from slope-area analysis is 4.37±0.37, which results in a $k_s$ value of 79.16±29.35. This indicates that the steepness indices will have large uncertainties even for high linear correlation of the log-transformed slope-area plot. Hence, the integral approach is much better for calculating channel steepness.

Concavity indices usually vary along river channels where different substrates outcrop (e.g. alluvium, and bedrock). For example, along the Juan River, we identified colluvial (log(slope) ~ -1, drainage area<0.16 km$^2$, elevation>700 m), bedrock (decreasing channel gradient) and alluvial (channel slope decreases in a much higher concavity, drainage area>8.89 km$^2$, elevation<150 m) channels from the log-transformed slope-area plot for their variable concavities (Figs. 4a and b). As shown in Fig. 4a, these channels are characterized by different concavities, consistent with estimates from Snyder et al (2000). According to the concavity of the bedrock portion of the river ($θ$=0.52, derived from the integral approach, Fig. 4c), the bedrock channel profile is converted into a straight line (Fig. 4d).

Nevertheless, for the integral approach, it is difficult to recognize bedrock and alluvial channels along a river profile. When computing χ-z plots ($A_{cr}$=0.16 km$^2$, for the whole fluvial channel including both bedrock and alluvial portions) based on a series of concavity values, the best fit $θ$ is 0.72 (Fig. 4e). As shown by the transformed profile (Fig. 4f), a knickpoint (at

elevation of ~400 m) occurs on the channel. Below the knickpoint, the alluvial and bedrock portions share the same slope ($k_s$=3354±20 m$^{1.44}$) despite the different channel substrates. Above the knickpoint, the $k_s$ value is 1667±15 m$^{1.44}$. Variations in the slope of χ-$z$ plot may be treated as spatially or temporarily variant rock uplift rates (Goren et al., 2014; Perron and Royden, 2013; Royden and Perron, 2013). However, no knickpoint occurs on stream Juan because the river has been controlled by uniform rock uplift and under steady state (Snyder et al., 2000; see Sect. 4.2 for discussion). Thus, a χ-$z$ plot generated by a single concavity may lead to misestimates in stream power parameters. We should recognize changes in concavities from slope-area space.

According to the log-transformed slope-area plots, we identified bedrock channels of the 15 streams. Concavity indices were then calculated via both slope-area analysis and the integral approach. As shown in Fig. 5, both methods yielded similar concavities. Based on a mean θ value of 0.45±0.10 (1σ), we computed χ-$z$ plots and normalized steepness indices ($k_{sn}$) with uncertainty estimates (Fig. 6). The uncertainties in steepness indices (no statistical test) are nearly lower than 1.0% (Fig. 6).

We run statistic tests (Durbin-Watson test and Spearman rank correlation coefficient test) for the integral approach and slope-area analysis. For the integral approach, all the DW statistics are lower than $D_L$ (Fig. 7a), indicating serially correlated residuals. Then, we revised the elevation and χ data according to Eq. (9) (Fig. 8). The DW statistics of revised χ-$z$ plots are all between $D_U$ and 4-$D_U$ (Fig. 7a), indicating independent residuals. The results of linear fit are shown in Fig. 8. The uncertainties in steepness indices (revised by Durbin-Watson test) are about 2.4% - 9.9% (Fig. 8), which are much higher than those without statistical test (lower than 1%, Fig. 6). In addition to uncertainty estimate, auto-correlated residuals can also bias the regression coefficient, steepness. The channel steepness values of streams Fourmile, Kinsey and Hardy are 57.01, 103.90 and 58.78 m$^{0.9}$ (Fig. 6). While revised by Durbin-Watson test, these values are 36.33, 82.16 and 76.65 m$^{0.9}$, respectively (Fig. 8). Then, steepness varies about 25.6% - 58.3% (dividing the difference of the two kinds of steepness indices by the values revised by Durbin-Watson test). Due to the influence of auto-correlated residuals on both the estimated value and precision of steepness, Durbin-Watson test is necessary when applying the integral approach. For slope-area analysis, the DW statistics are all between $D_U$ and 4-$D_U$ (Fig. 7b), showing no auto-correlation.

We also calculated t-statistics for both slope-area analysis and the integral approach (Figs. 7a and b). All the results are less than 2, indicating homoscedastic residuals. Despite no heteroscedasticity found in our study area, we suggest that Spearman rank correlation coefficient test should also be done because the test is a part of linear regression (statistically).

In addition to statistic tests, another way proposed by Perron and Royden (2013) to estimate uncertainty in steepness is to make multiple independent calculations of different river profiles. From Fig. 6, the mean $k_{sn}$ of high uplift zone ($U$=4 mm/yr) is 104.40±14.06, and that of low uplift zone ($U$=0.5 mm/yr) is 71.25±10.08. The standard errors of the mean $k_{sn}$ among profiles are considerably larger than that for individual streams. However, for multiple profiles under similar geological and/or climatic settings, this approach should provide more meaningful estimates of uncertainty.

## 4 Discussion

Even though it gives highly uncertain channel steepness values, slope-area plots make no assumptions about $\theta$ and therefore are more sensitive than $\chi$ analysis for detecting spatially varying concavities. Slope-area analysis, thus, is useful to identify difference in substrates along a river (e.g. bedrock, alluvium), which can be used as regression limits to apply the integral method. The integral approach yields better-constrained values of $\theta$ and $k_{sn}$. Combining these methods with statistical tests provides more reliable results while applied to perform stream profile analysis. In the following sections, we will discuss the parameter uncertainty and steady assumption to better illustrate this method.

### 4.1 Uncertainty of channel concavity

Perron and Royden (2013) considered that the uncertainty in channel concavity derived from a linear regression of the log-transformed slope–area plot described how precisely one can measure the slope of the plot, not how precisely the parameter is known for a given landscape. They suggested that the difference between $\theta$ values that best linearize the main stem profile and that maximize the co-linearity of the main stem with its tributaries could be an estimate of uncertainty in $\theta$ for an individual drainage basin.

In most cases, the $\theta$ value that collapses the main stem and its tributaries is often used as a reference concavity (Mudd et al., 2014; Perron and Royden, 2013; Willett et al., 2014; Yang et al., 2015). In fact, supposing a drainage basin under uniform geologic and climatic settings, this kind of $\theta$ value can be compared with the mean value of concavities of the stem and its tributaries. We thereafter, name these two concavities $\theta_{Co}$ (derived from the collinearity test) and $\theta_{mR}$ (from averaging the concavity values of all the streams within a catchment), respectively.

We extracted the stems and tributaries of streams, Singley, Davis, Fourmile and Cooskie (Fig. 9a), based on $A_{cr}$ of 0.1-0.16 km$^2$ (Fig. 5). We calculated the correlation coefficients of $\chi$-$z$ plots based on a range of $\theta$ (Figs. 9b-d). The $\theta_{mR}$ of catchments are 0.45, 0.48, 0.43, and 0.55. We also derived $\theta_{Co}$ which collapses the stem and tributaries, 0.45, 0.45, 0.45, and 0.55 (Fig. 10). Both $\theta_{Co}$ and $\theta_{mR}$ are similar to the stem concavities, 0.50, 0.42, 0.50, and 0.45 (Fig. 5). Hence, for steady state bedrock channels under uniform lithologic and climatic settings, all three kinds of concavities should be similar. Then, the difference between these $\theta$ values could be an estimate of concavity uncertainty.

However, concavity varies in streams consisting of both bedrock and alluvial channels. We extracted the stems and tributaries of streams, Hardy, Juan, Howard and Dehaven (Fig. 11a). The $\theta_{mR}$ values of them are 0.57, 0.68, 0.73, and 0.73 (Figs. 11b-e), similar to the stem concavities (0.63, 0.70, 0.72, and 0.75) (Figs. 11b-e), but larger than the $\theta_{Co}$ (0.45, 0.45, 0.45, and 0.55) (Fig. 12). In such case, differences between $\theta_{Co}$ and $\theta_{mR}$ are not random errors and cannot be estimates of concavity uncertainty.

Nevertheless, $\theta_{Co}$ values (0.45, 0.45, 0.45, and 0.55) are similar to the concavities of bedrock reaches of stems (0.55, 0.52, 0.55, and 0.40) (Fig. 5). Then, the differences between $\theta_{Co}$ and concavities of bedrock reaches may be estimates of

uncertainties in $\theta$. Hence, the reference concavity collapsing the stem and its tributaries works well even for all the profile data consisting of both bedrock and alluvial channels.

In most cases, a somewhat higher constant critical area (e.g. 1 or 5 km²) is assumed to calculate $\chi$ values of fluvial channels (Goren et al., 2014, 2015; Willett et al., 2014; Yang et al., 2015). Here we extracted streams of four drainages (Fig. 13a), Hardy, Juan, Howard, and Dehaven, based on a critical area of 0.5 km² (three or four times the actual values). We then derived the concavities that best linearize stems (0.73, 0.78, 0.82, and 0.84) (Figs. 13b-e), $\theta_{mR}$ (0.60, 0.80, 0.75, and 0.75) (Figs. 13b-e), and $\theta_{Co}$ (0.40, 0.50, 0.45, and 0.55) (Fig. 14), respectively. All the results are similar to those based on actual critical areas (Figs. 11 and 12). Hence, choosing a uniform $A_{cr}$ somewhat different to the actual values might be reasonable and would not have significant influence.

## 4.2 Steady state assumption of streams in the MTJ region

Usually, river shape may not be diagnostic of equilibrium conditions. In some places, recent work on inversion of drainage patterns for uplift rate histories indicates that river profile shapes are controlled by spatio-temporal variations in uplift rate moderated by erosional processes (Pritchard et al., 2009; Roberts and White, 2010; Roberts et al., 2012).

In the MTJ region, the uplift rates determined by marine terraces are variable in space and time (0-4 mm/a, Merritts and Bull, 1989). However, in the low-uplift zone (streams Hardy to Dehaven), uplift rates have been approximately constant for at least 0.33 Ma (Merritts and Bull, 1989). The bedrock-channel reaches are probably not affected by sea-level fluctuations (Snyder et al., 2000). These streams thus can be in or near equilibrium. Nevertheless, disequilibrium conditions are likely in regions of high-uplift rate (e.g. the rivers north of 40°N). To test the steady state assumption, we modelled the uplift rate histories.

Erosional parameters in the stream power model (e.g., $m$ and $n$) and uplift histories can be determined from joint inversion of drainage network (Glotzbach 2015; Goren et al., 2014; Pritchard et al., 2009; Rudge et al., 2015). Here, we utilized the method of Goren et al (2014). For spatially variant rock uplift, the study area is divided into four distinct zones, from north to south, the north transition zone (streams Singley to Cooskie), the King Range high-uplift zone (streams Randall to Buck), the intermediate-uplift zone (stream Horse Mtn), and the low-uplift zone (streams Hardy to Dehaven) (Figs. 15a-d; Snyder et al., 2000). Within each zone, we assumed spatially invariant rock uplift for small drainage areas and similar uplift rates determined from marine terraces (Merritts and Bull, 1989). Snyder et al (2000) suggested $n \sim 1$ and variable $K$ between the high- and low-uplift zones. According to the linear inversion model of Goren et al (2014), the present river channel elevation is determined by both rock uplift rate and response time, $\tau(x)$ (time for perturbations propagating from the river outlet, at $x=0$, to a point $x$ along the channel):

$$z(x) = \int_{-\chi(x)}^{0} U^*(t^*)\, dt^* \tag{12}$$

$$U^* = U/(KA_0^m), \quad t^* = KA_0^m t \tag{13}$$

For the linear model ($n=1$) and $A_0=1$ m$^2$, response time $\tau(x)=\chi(x)/K$. The scaled time $t^*$ has the same unit of $\chi$, and $U^*$ is dimensionless rock uplift rate.

Since $\chi$-$z$ plot may be affected by other factors (e.g., climate and lithology), we extracted all the fluvial channels and calculated a mean $\chi$-$z$ plot for each zone (Figs. 15e-h). We defined $z_1$, $z_2$, ..., $z_N$ and $\chi_1$, $\chi_2$, ..., $\chi_N$ to be the elevations and $\chi$ values of $N$ data points along a fluvial channels network ($N=10$ here). Then, based on Eq. (12), the dimensionless rock uplift histories for the four zones are shown in Fig. 15i. For the low-uplift zone, alluvial channels in the lower reaches were excluded for them being affected by sea-level fluctuations.

We utilized variable erodibility ($K=U/k_{sn}$) values to calculate rock uplift rates. The $K$ values for transition, high-uplift, intermediate and low-uplift zones are $6.17\times10^{-5}$ m$^{0.1}$/a, $3.82\times10^{-5}$ m$^{0.1}$/a, $3.38\times10^{-5}$ m$^{0.1}$/a, and $0.37\times10^{-5}$ m$^{0.1}$/a, respectively. According to the inferred uplift histories (Fig. 15j), the maximum response time (the perturbations migrating from the river outlet to water head) differs significantly from low- (0.43 Ma) to high-uplift (0.16 Ma) zones. The rock uplift rates in the low- and intermediate-uplift zones have been constant (~0.3-0.4 mm/a since 0.4 Ma, and ~2-2.5 mm/a since 0.16 Ma, respectively). The north transition and high-uplift zones both experienced increases in the uplift rates (from ~2.5 mm/a to 3.3 mm/a, and from ~3.7 mm/a to 4.3 mm/a, respectively) starting about 0.12 Ma ago. However, the increase ratios are much lower. Considering the maximum response time (~0.16 Ma), the uplift rates have been constant for a relatively long period. In addition, no large knickpoints are found along the rivers. All of these indicate that the rivers have been reshaped by the recent tectonic activities and have reached steady state.

In the recent 0.02 Ma, the rock uplift rates seem to be a bit lower (Fig. 15j). That may be due to variant channel concavities. The reaches downstream are usually characterized by rapidly decreasing gradient (higher concavities). Then, lower $U^*$ will be produced when using a reference concavity (0.45). As a result, the modelled rock uplift rates will be low. The variance in channel concavity may indicate difference in river substrate (e.g. sedimentation affected by sea-level fluctuations) rather than tectonics (Snyder et al., 2000).

## 4.3 Influence of elevation data uncertainty

Roberts et al (2012) noticed that the slope-area methodology might produce unstable results because small amounts of randomly distributed noise added to river profile will cause significant change in channel gradient. In spite of little knowledge about the elevation data uncertainty here, we utilized different datasets and various data handling methods (data smoothing and sampling) to calculate channel slope with different uncertainties. Then, to some extent, the influence of data uncertainty can be tested.

In the analysis above, the channel slope is derived from 1 arc-second SRTM DEM via 300 m smoothing window and 20 m contour sampling interval. Then, we reanalysed the streams in high- and low-uplift zones based on 1/3 arc-second USGS DEM (downloaded from https://catalog.data.gov/dataset/national-elevation-dataset-ned-1-3-arc-second-downloadable-data-collection-national-geospatial). We calculated the channel slope via 300 m smoothing window and 20 m contour sampling interval (Figs. 16a and d), 300 m smoothing window and 10 m contour sampling interval (Figs. 16b and e), and 100 m

smoothing window and 10 m contour sampling interval (Figs. 16c and f), respectively. To get average values, slope-area data from all the streams within the same zone were composited.

We chose 0.1-3 km$^2$ as regression limits for the high-uplift zone and 0.2-8 km$^2$ for the low-uplift area. The channel concavity and steepness ($k_{sn}$) were calculated by linear regressing the log-transformed slope-area data and χ-z plots ($\theta_{ref}$=0.45), respectively. The stream concavity indices in the high-uplift zone (0.41±0.05) and low-uplift region(0.48±0.03) are similar to or within error of the estimates reported by this study (0.45±0.10, 1 Arc-second SRTM DEM), Wobus et al (2006) (0.57±0.05, 10-m-pixel USGS DEM), and Snyder et al (2000) (0.43±0.11, 30 m USGS DEM). All the error estimates are characterized by 1σ. Mean $k_{sn}$ values of 109 and 60 m$^{0.9}$ in the high- and low-uplift zones, respectively, yield a ratio of $k_{sn}$(high)/$k_{sn}$(low) of ~1.82, which mirrors the findings of both Snyder et al (2000) and Wobus et al (2006). We find no distinct difference in concavity and channel steepness indices when using different datasets and data handling methods.

Utilising different datasets may cause some differences in parameter estimate for an individual catchment. For example, when using the integral approach, the resulting channel concavity of stream Cooskie is 0.45 (in Sect. 3) (1 arc-second SRTM DEM) but 0.36 in Perron and Royden (2013) (1/3 arc-second USGS DEM). However, for averaged results (as done in Sect. 4.3), uncertainty in elevation data may not cause distinct differences in parameter estimates in this study area (e.g. θ and $k_{sn}$).

## 4.4 Disequilibrium circumstances in large rivers

The case study has disadvantages of including only short (<10 km long; < 20 km$^2$ area) and steady streams. In many landscapes, especially large rivers, this steady assumption will not be met (Harkins et al., 2007; Wobus et al., 2006; Yang et al., 2014). To explore the effect of landscape transience, we analyzed Mattole River, a large river in the MTJ region (Fig. 17a). Here, 1/3 arc-second USGS DEM was used.

Using a 300 m smoothing window and 20 m contour sampling interval, we derived a log-transformed slope-area plot of the stem (Fig. 17b). We recognized the critical threshold of drainage area, $A_{cr}$, ~0.1 km$^2$ and at the elevation of ~450 m (Figs. 17b and c) from the slope-area plot by eye. A knickpoint was detected by the scaling break in the slope-area data and then marked in the shaded-relief map (Fig. 17a) and the river profile (Fig. 17c). The knickpoint is located at an elevation of ~280 m. The concavity indices above (0.61±0.01) and below (0.58±0.07) the knickpoint are nearly the same. To compare with the adjacent streams, a reference concavity $\theta_{ref}$=0.45 was used to calculate the channel steepness. Using the integral approach and two statistic tests, we derived the $k_{sn}$ above (10.81±0.86 m$^{0.9}$) and below (17.44±1.16 m$^{0.9}$) the knickpoint. However, in the adjacent streams, (e.g. Davis, Fourmile), the $k_{sn}$ values are much larger than 60 m$^{0.9}$. In addition to spatial variations in Holocene uplift rates of marine platforms (Merritts, 1996), these steepness indices suggest that other variables (e.g. sediment flux and lithology) may affect channel steepness. This might limit our ability to quantitatively relate steepness indices to uplift rates in this field setting, as noticed by Wobus et al (2006).

Usually, the method of best linearizing χ-z plot is used to compute θ for a steady state bedrock river profile (Perron and Royden, 2013). However, channel may be transient in which case previous authors have suggested either segmentation of χ profiles (Mudd et al., 2014) or interpretation through inversion methods (e.g., Goren et al., 2014). We computed the

correlation coefficients between the channel elevation and $\chi$ values ($A_{cr}$=0.1 km$^2$, $A_0$=1 m$^2$) of the stem based on a range of $\theta$ (Fig. 17d). The best linear fit corresponds to $\theta$=0.30 ($k_s$=1.45±0.06 m$^{0.6}$, R=0.985, Fig. 17e), which is distinctly different from the result of slope-area analysis.

We extracted all the tributaries of the Mattole river and calculated their $\chi$-$z$ plots based on a range of $\theta$ values (Figs. 18a-d). The elevation scatters of the $\chi$-$z$ plots are plotted against $\theta$ values (Fig. 18e). The $\theta$ value that collapses the main stem and its tributaries is 0.45, showing the reasonability of using 0.45 as a reference concavity to calculate the stem $k_{sn}$. As shown by Fig. 18c, the knickpoint (with an elevation of about 280 m) can also be detected from the $\chi$-$z$ plot of the stem. Both the slope-area data (Fig. 17b) and the $\chi$-$z$ plot based on a $\theta$ value derived from collinearity test (Fig. 18c) detect the unsteady signal on the trunk stream of the Mattole river, despite the best linearity for the integral approach (Fig. 17e). Then, we can find that a river may be in disequilibrium condition in spite of a linear relationship in the $\chi$-$z$ plot. In some cases, uplift can be inserted along rivers, which makes values of $\chi$ difficult to interpret (Czarnota et al.,2014;Paul et al., 2014;Pritchard et al., 2009; Roberts and White, 2010; Roberts et al., 2012; Wilson et al., 2014).

Based on $\theta_{ref}$=0.45, we calculated the map of channel steepness with an elevation interval of 100 m. The channel steepness values range from 1 to 273. As shown in Fig. 19, the lower $k_{sn}$ values are along the whole stem and its tributaries (low elevation) above the knickpoint while higher values are along the upstream (high elevation) of tributaries below the knickpoint. Among the tributaries in the west of the stem, channel steepness decreases from the central part (near streams Big to Shipman, high-uplift zone) towards both north (close to stream Fourmile, north transition zone) and south (near streams Horse Mtn and Telegraph, intermediate-uplift zone). Both the spatial pattern of $k_{sn}$ and the positive relationship between $k_{sn}$ and elevation may indicate a tectonic control on channel steepness despite other potential variables.

## 5Conclusion

In this contribution, we coupled the advantages of slope-area analysis and the integral approach to steady state bedrock river profile analysis. First, we identified colluvial, bedrock and alluvial channels from a log-transformed slope-area plot. Utilizing the integral approach, we then derived concavity and steepness indices of a bedrock channel. Finally, via Durbin-Watson statistic test, we examined and eliminated serial correlation of linear regression residuals, which produced more reliable and robust estimates of uncertainties in stream power parameters.

### Acknowledgement

We thank Dr. Eric Kirby (Oregon State University) and Dr. Liran Goren (Ben-Gurion University of the Negev) for significant suggestions on slope-area analysis and the integral approach, respectively. We thank Amanda McDowell (Oregon State University) and Qi Ou (University of Cambridge) for revision in English writing. We are grateful for the grants from the National Science Foundation of China (41622204, 41272215, 41272196, 41590861, 41661134011), State Key Laboratory of Earthquake Dynamics (LED2014A03) and the Strategic Priority Research Program of the Chinese Academy of Sciences (XDB03020200). YW is also funded by state special supporting plan.

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

Table 1. Range of DW statistic and the related meaning

| DW Statistic | Meaning |
|---|---|
| $0 \leq DW \leq D_L*$ | Positively auto-correlated residuals |
| $D_L < DW \leq D_U*$ | Beyond the suitability of Durbin-Watson test |
| $D_U < DW < 4-D_U$ | Mutually independent residuals |
| $4-D_U \leq DW < 4-D_L$ | Beyond the suitability of Durbin-Watson test |
| $4-D_L \leq DW \leq 4$ | Negatively auto-correlated residuals |

*$D_L$ and $D_U$ represent the critical value of Durbin-Watson test and can be found in Durbin and Watson (1950).

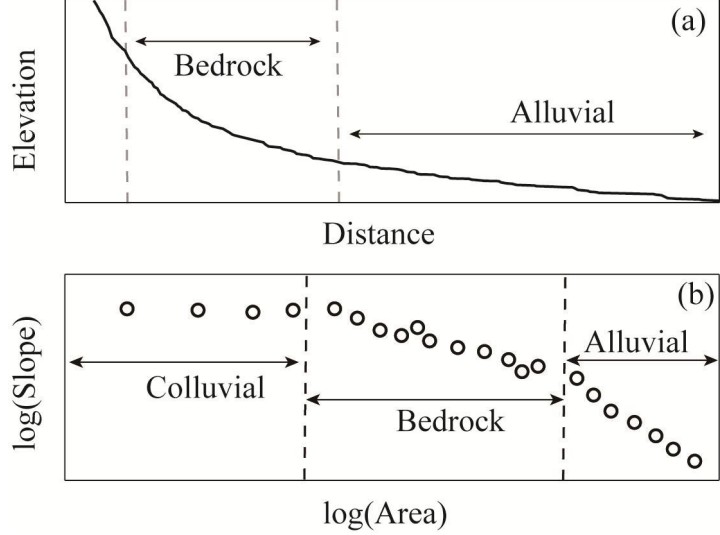

Figure 1.  Schematic of a steady state river profile consisting of colluvial, bedrock and alluvial channels, revised from Figures 7A and B in Snyder et al (2000). (a) Stream profile. (b) Log-transformed slope-area plot.

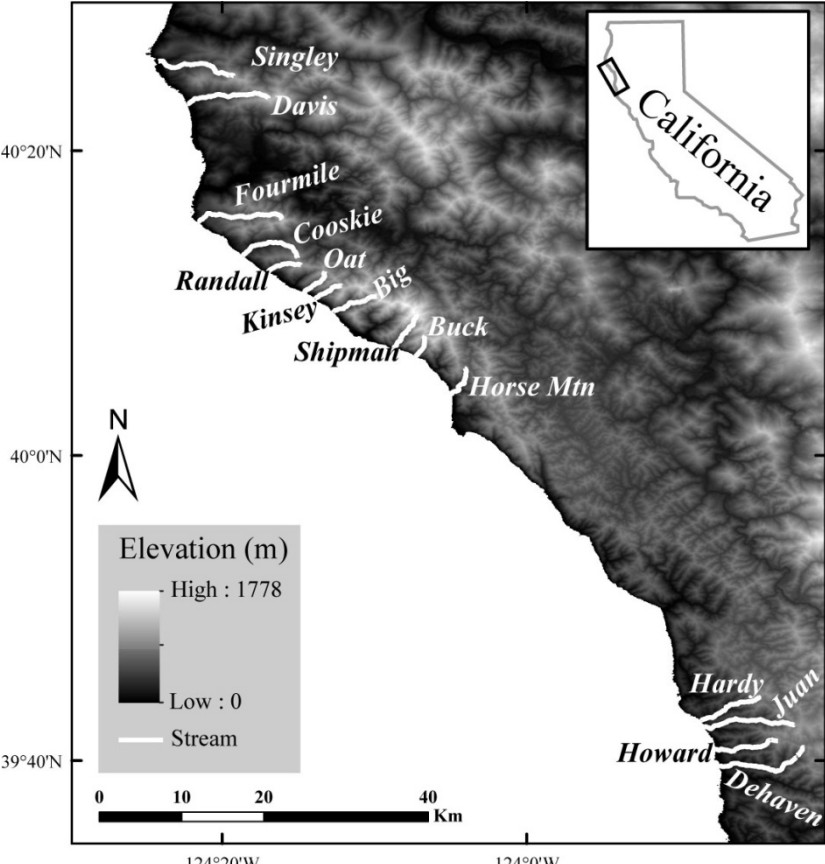

Figure 2. Streams in the Mendocino Triple Junction (MTJ) region of northern California, USA. Streams are from Snyder et al (2000). The elevation data are from 1 Arc-Second SRTM (http://earthexplorer.usgs.gov/)

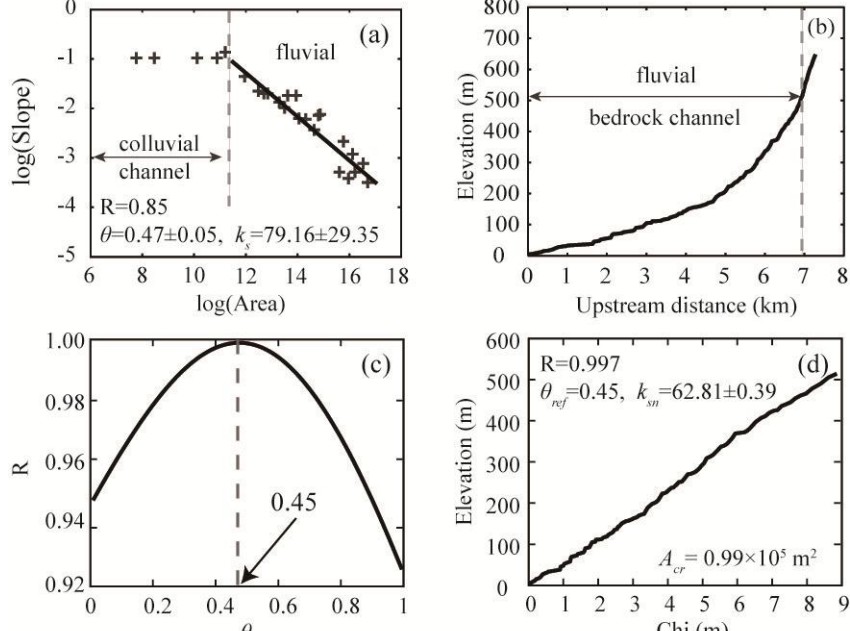

Figure3. Stream profile analysis of Cooskie. (a) Log-transformed slope-area plot. The slope was derived from the smoothed (horizontal distance of 300m) and re-sampled (elevation interval of 20m) elevation data. (b) The full river profile (without any smoothing or re-sampling) of Cooskie. (c) The correlation coefficients, R, as a function of $\theta$ for least-squares regression based on Eq. (5). The maximum value of R, which corresponds to the best linear fit, occurs at $\theta$=0.45 (dotted line and black arrow). (d) $\chi$-$z$ plot of the bedrock channel profile, transformed according to Eq. (3) with $\theta$=0.45, $A_{cr}$=0.1 km$^2$, and $A_0$=1 m$^2$.

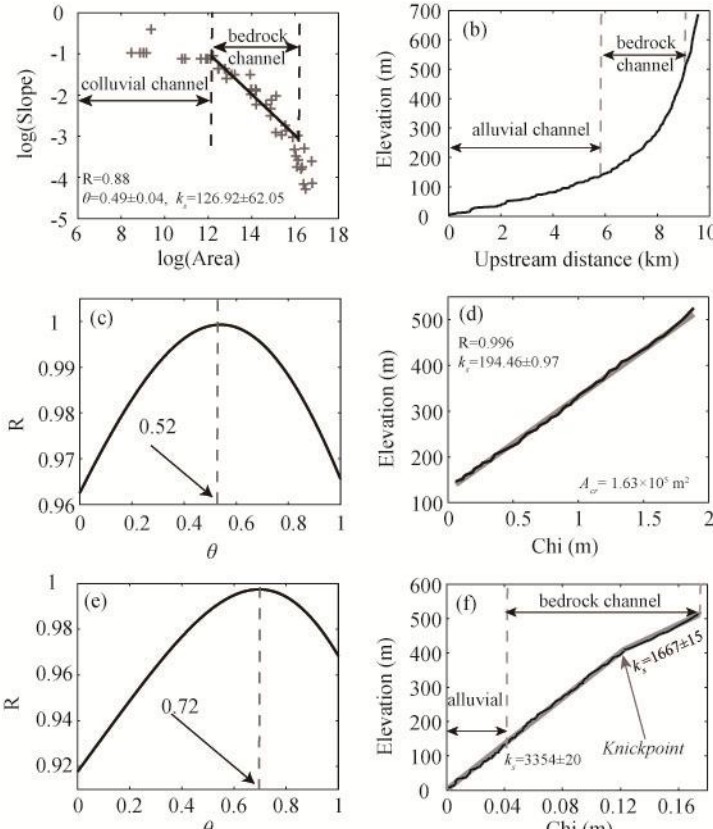

Figure 4. Stream profile analysis of Juan.(a) Log-transformed slope-area plot. The slope was derived from the smoothed (horizontal distance of 300m) and re-sampled (elevation interval of 20m) elevation data. (b) The full river profile (without any smoothing or re-sampling) of Juan. (c) The correlation coefficients of χ-z plots as a function of $\theta$ for the bedrock portion of the river. The maximum value of R occurs at $\theta$=0.52. (d) χ-z plot of the bedrock channel profile based on a concavity value of 0.52. (e) The correlation coefficients of χ-z plots as a function of $\theta$ for fluvial (both bedrock and alluvial) channel. The maximum value of R occurs at $\theta$=0.72. (f) χ-z plot of the fluvial (both bedrock and alluvial) channel profile based on a concavity value of 0.72.

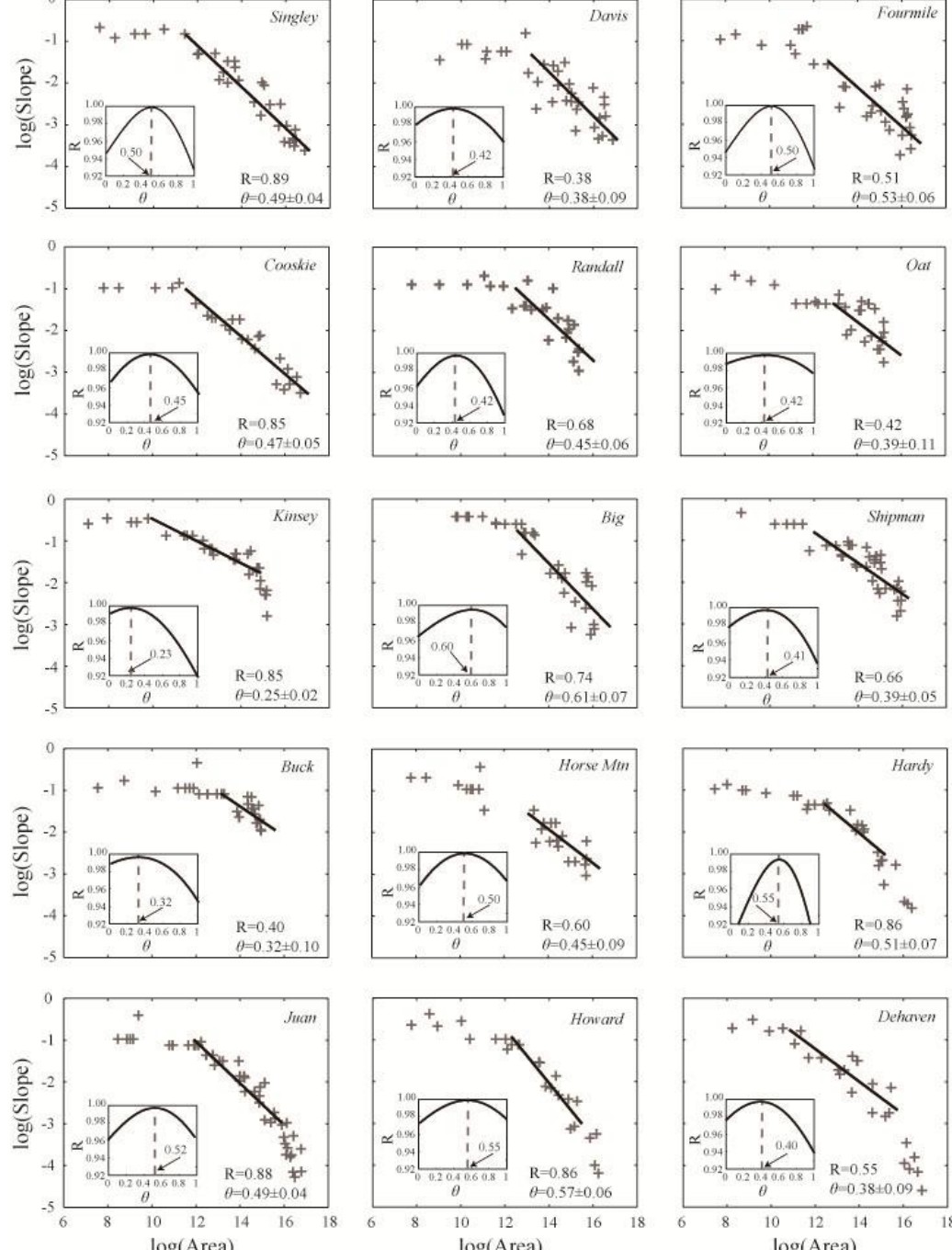

Figure5. Correlation coefficients derived from slope-area analysis and the integral approach. The slope was derived from the smoothed (horizontal distance of 300m) and re-sampled (elevation interval of 20m) elevation data. The correlation coefficients of χ-z plots as a function of θ for bedrock channels are shown in the left bottom. Then mean θ value is 0.45.

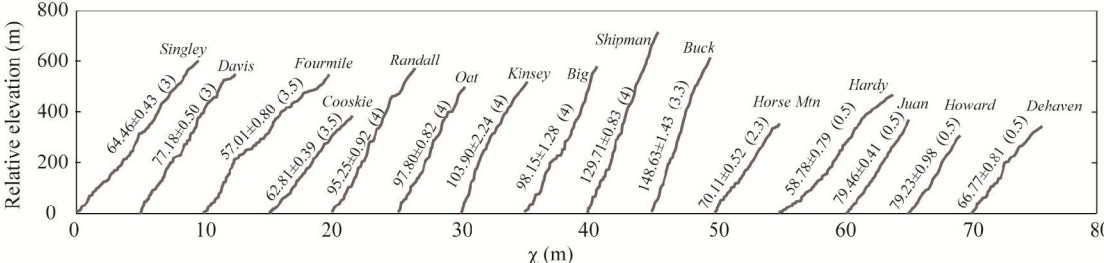

Figure 6.χ-z plots of the streams (bedrock channels) based on the mean concavity (0.45). Numbers are normalized channel steepness, $k_{sn}$, with the uncertainty estimates. Numbers in the parentheses are uplift rates with a unit of millimetre per year (Snyder et al., 2000). Italic characters are stream names.

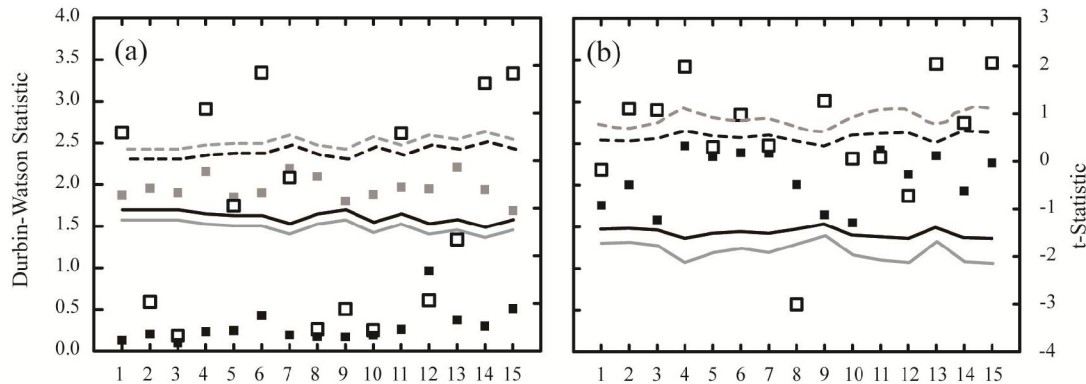

Figure 7. Statistic tests for the integral approach (a) and slope-area analysis (b). Black hollow squares are t-statistic of Spearman rank correlation coefficient. Black solid squares are Durbin-Watson statistics. The gray solid line, black solid line, black dashed line and gray dashed line are $D_L$, $D_U$, 4-$D_u$, and 4-$D_L$. Gray squares in Figure (a) are the Durbin-Watson statistics of revised χ-z plots. The river numbers

10    1 to 15 indicate streams: Singley, Davis, Fourmile, Cooskie, Randall, Oat, Kinsey, Big, Shipman, Buck, Horse Mtn, Hardy, Juan, Howard, and Dehaven.

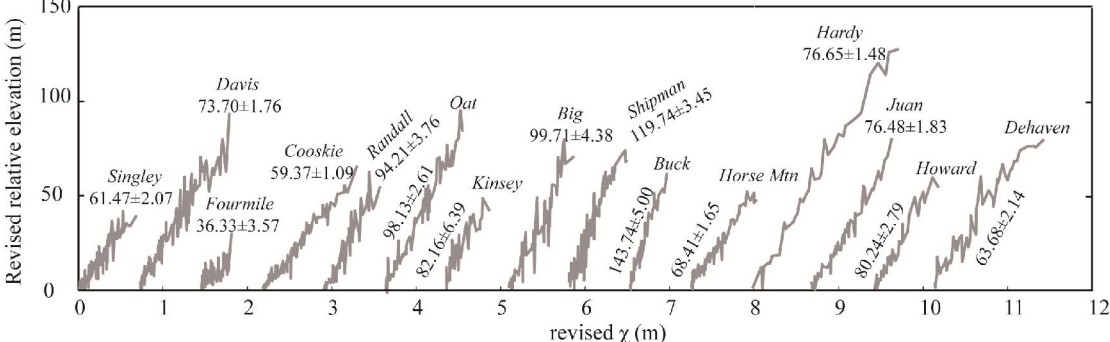

Figure 8. Revised relative elevation and χ values. Gray lines and Numbers are revised data and steepness index with uncertainty estimates.

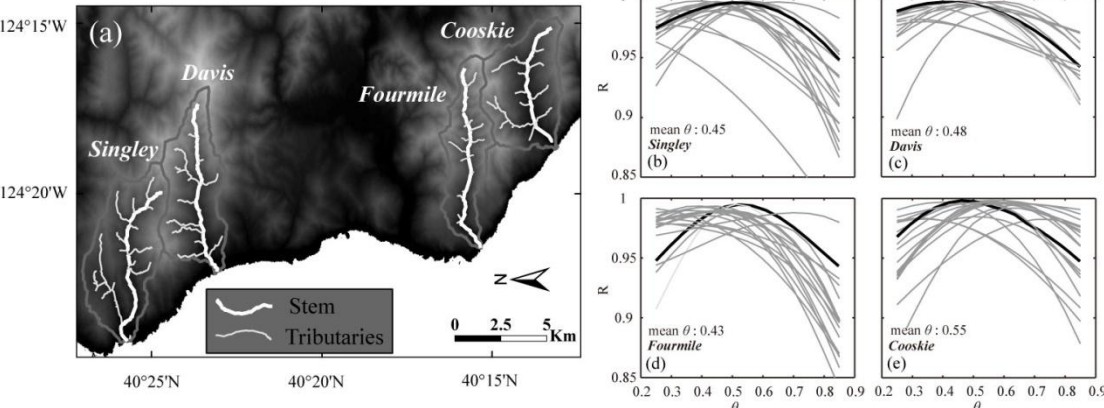

Figure9. Correlation coefficients of χ-z plots as a function of θ. (a) Location of the streams. (b-e) Correlation coefficients of χ-z plots based on a range of θ values. Black thick lines indicate stems.

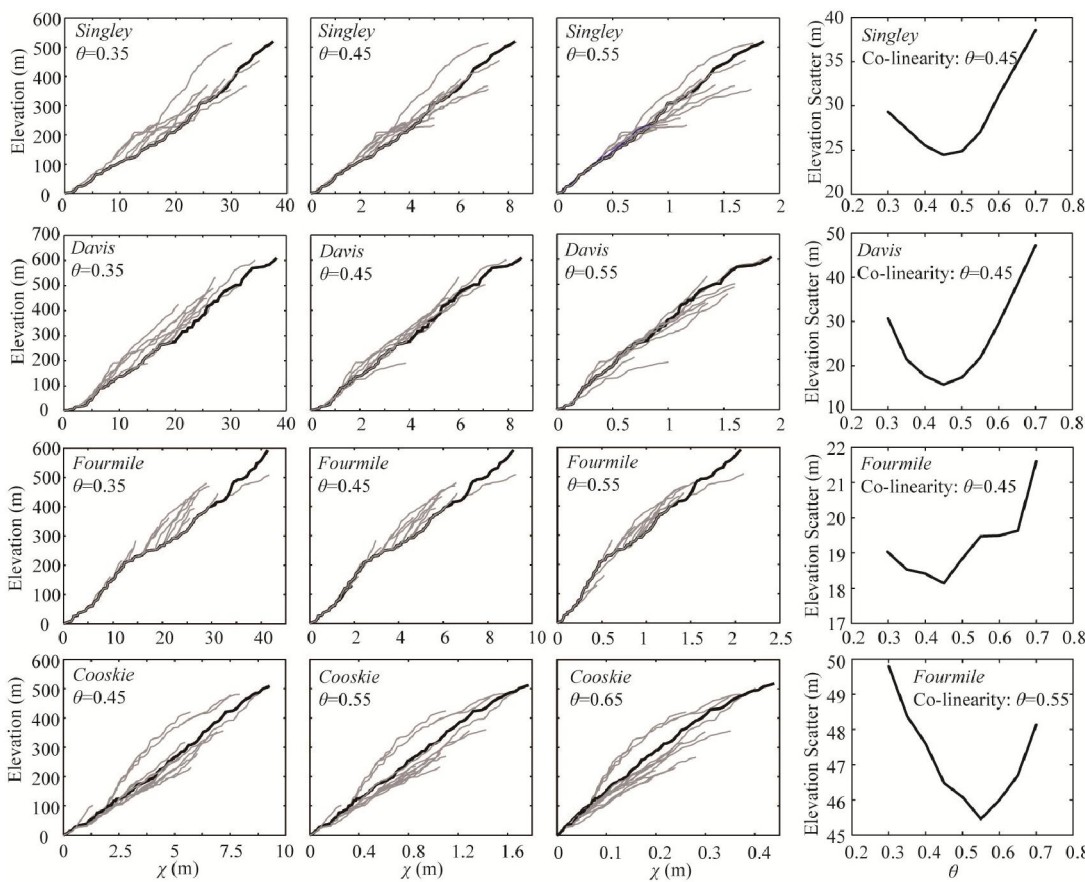

Figure10 Concavity values that maximize the co-linearity of the main stem with its tributaries. Black thick lines in the χ-z plots are stems.

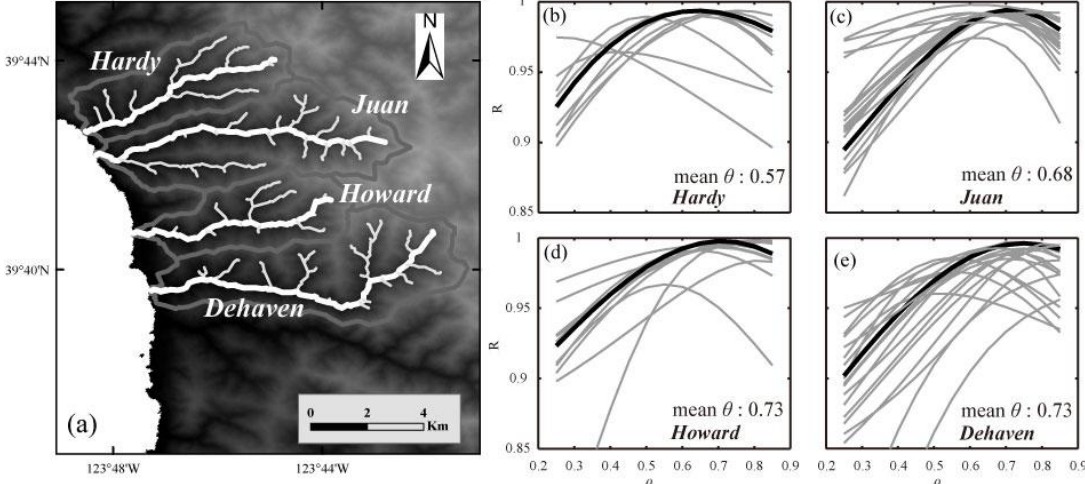

Figure11. Correlation coefficients of χ-z plots as a function of θ. (a) Location of the streams. (b-e) Correlation coefficients of χ-z plots based on a range of θ values. Black thick lines indicate stems.

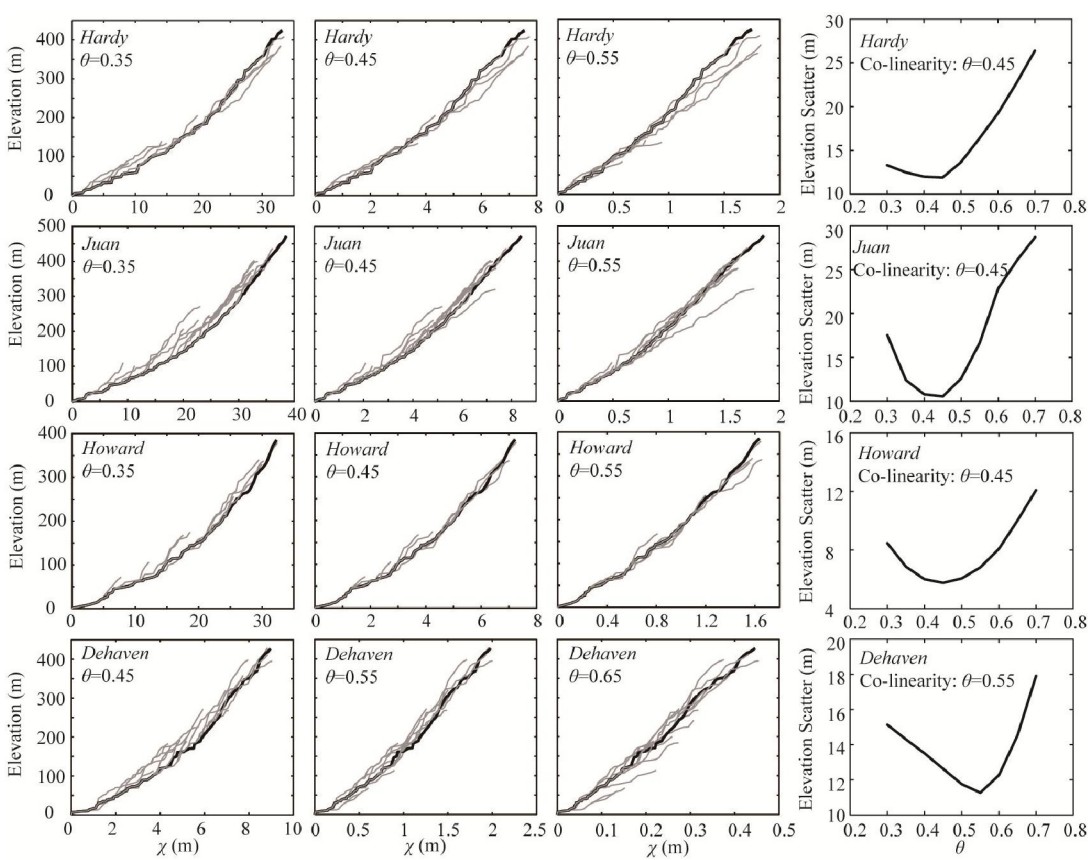

Figure12. Concavity values that maximize the co-linearity of the main stem with its tributaries. Black thick lines in the χ-z plots are stems.

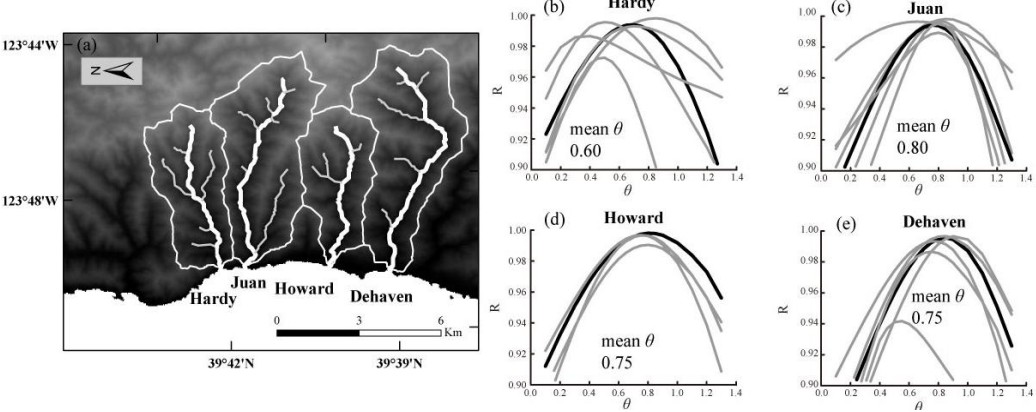

Figure13. Correlation coefficients of χ-z plots as a function of θ. (a) Location of the streams. Streams are extracted with a critical area of 0.5km². (b-e) Correlation coefficients of χ-z plots based on a range of θ values. Black thick lines indicate stems.

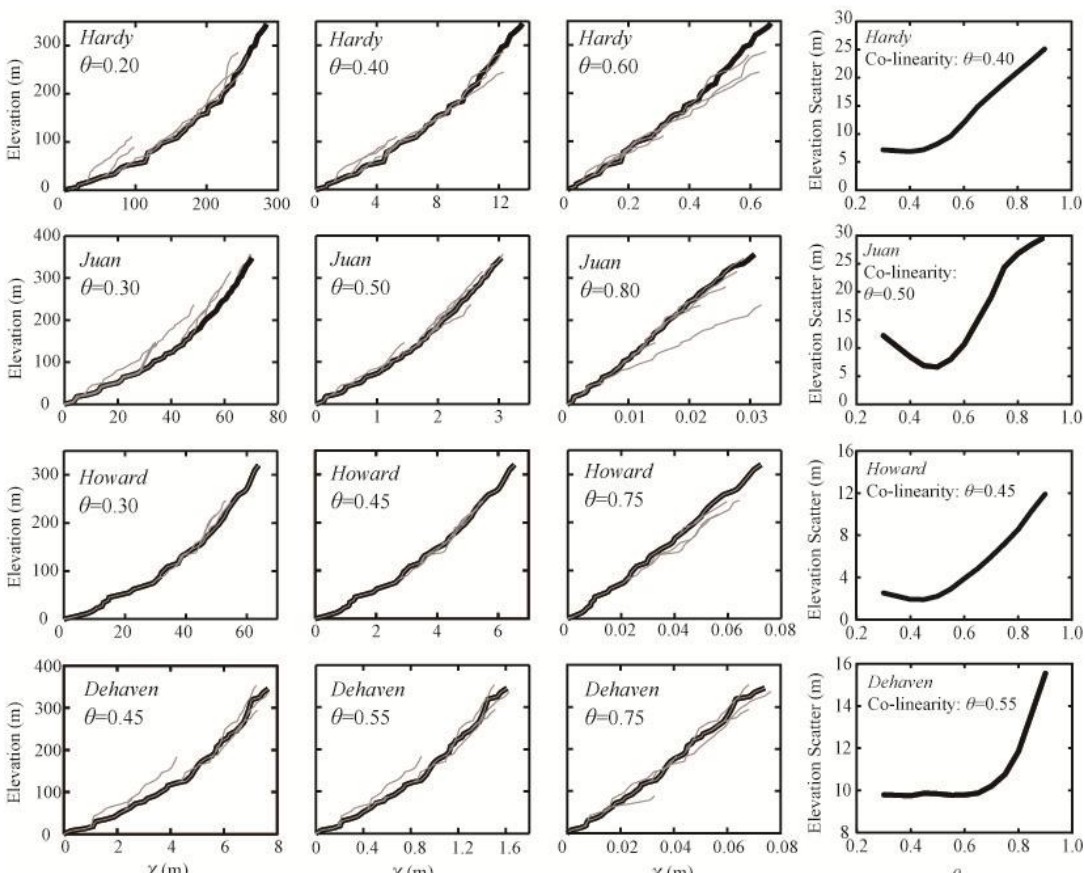

Figure14. Concavity values that maximize the co-linearity of the main stem with its tributaries. Black thick lines in the χ-z plots are stems.

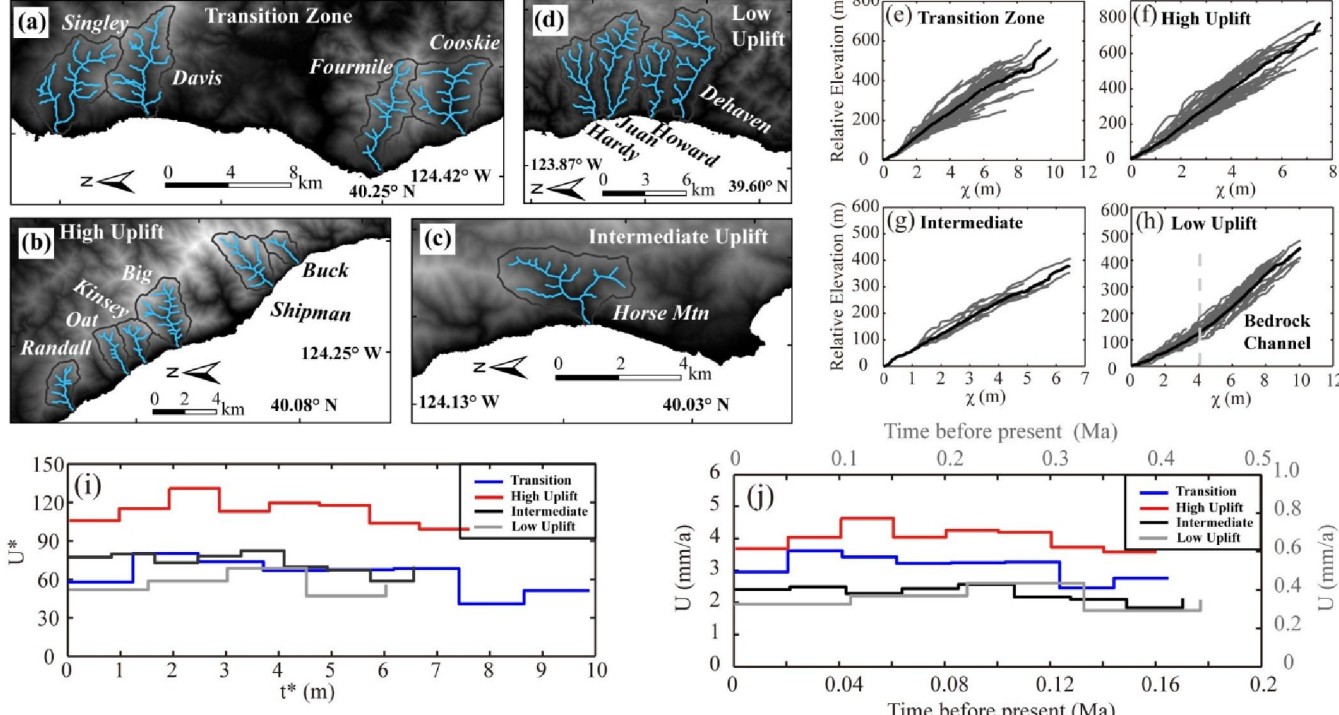

Figure 15. Uplift histories inferred from the stream profiles. (a)-(d) The map of streams in the four zones: north transition zone (a), King Range high-uplift zone (b), intermediate-uplift zone (c), and low-uplift zone (d). (e)-(h) The $\chi$-$z$ plots of the streams within each zone ($A_0$=1 m$^2$). The black line indicates an average result. (i) Scaled U*as a function of scaled time t*. (j) Inferred relative uplift rate as a function of time before the present. The left-bottom black axes show the results of north transition, high-uplift and intermediate-uplift zones. The right-top gray axes show the result of low-uplift zone.

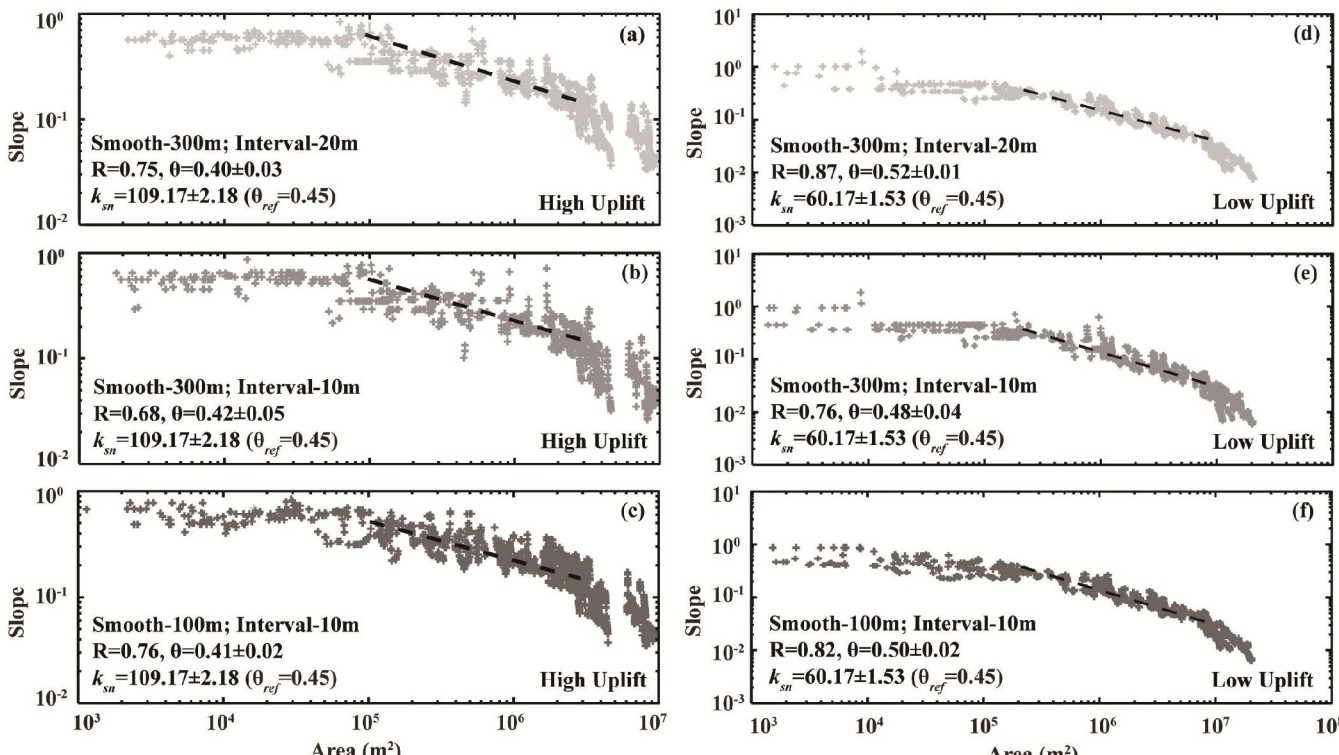

Figure 16. Log-transformed slope-area plots of streams in the high-uplift (a-c) and low-uplift (d-f) zones. Streams within the same zone are composited. The slope data are calculated via different methods: 300 m smoothing window and 20 m contour sampling interval (a and d), 300 m smoothing window and 10 m contour sampling interval (b and e), and 100 m smoothing window and 10 m contour sampling interval (c and f). Elevation data are from 1/3 arc-second USGS DEM (downloaded from https://catalog.data.gov/dataset/national-elevation-dataset-ned-1-3-arc-second-downloadable-data-collection-national-geospatial).

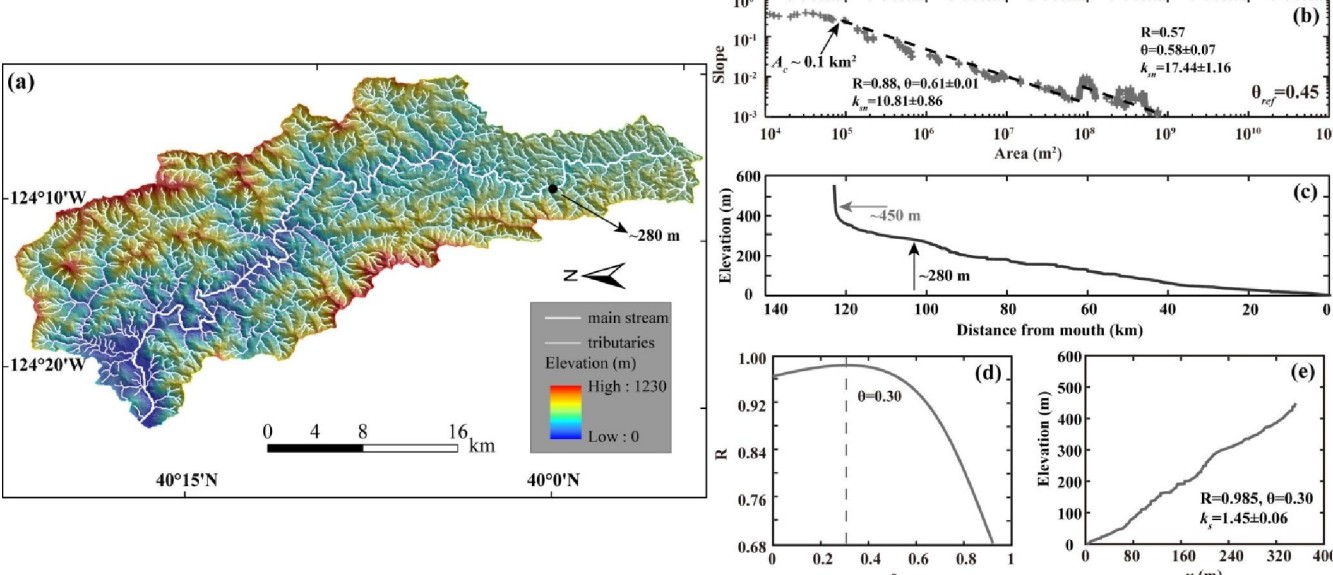

Figure 17. The slope-area data and χ-*z* plot of the stem of Mattole river. (a) Map of the stem and its tributaries in the Mattole drainage basin. Elevation data are from 1/3 arc-second USGS DEM (downloaded from https://catalog.data.gov/dataset/national-elevation-dataset-ned-1-3-arc-second-downloadable-data-collection-national-geospatial). (b) The log-transformed slope-area plot of the stem (300 m smoothing window and 20 m contour sampling interval). A knickpoint is detected from the plot with variant $k_{sn}$ along the channel. (c) River profile of the stem. The gray arrow indicates the dividing point (~450 m) between the colluvial and fluvial portions. The black arrow shows the knickpoint (~ 280 m) on the stem. (d) The correlation coefficients between elevation and χ values as a function of $\theta$. The maximum value of R, which corresponds to the best linear fit, occurs at $\theta$=0.30 (gray dashed line). (e) The χ-*z* plot of the stem, transformed according to Eq. (3) with $\theta$=0.30, $A_{cr}$=0.1 km$^2$, and $A_0$=1 m$^2$.

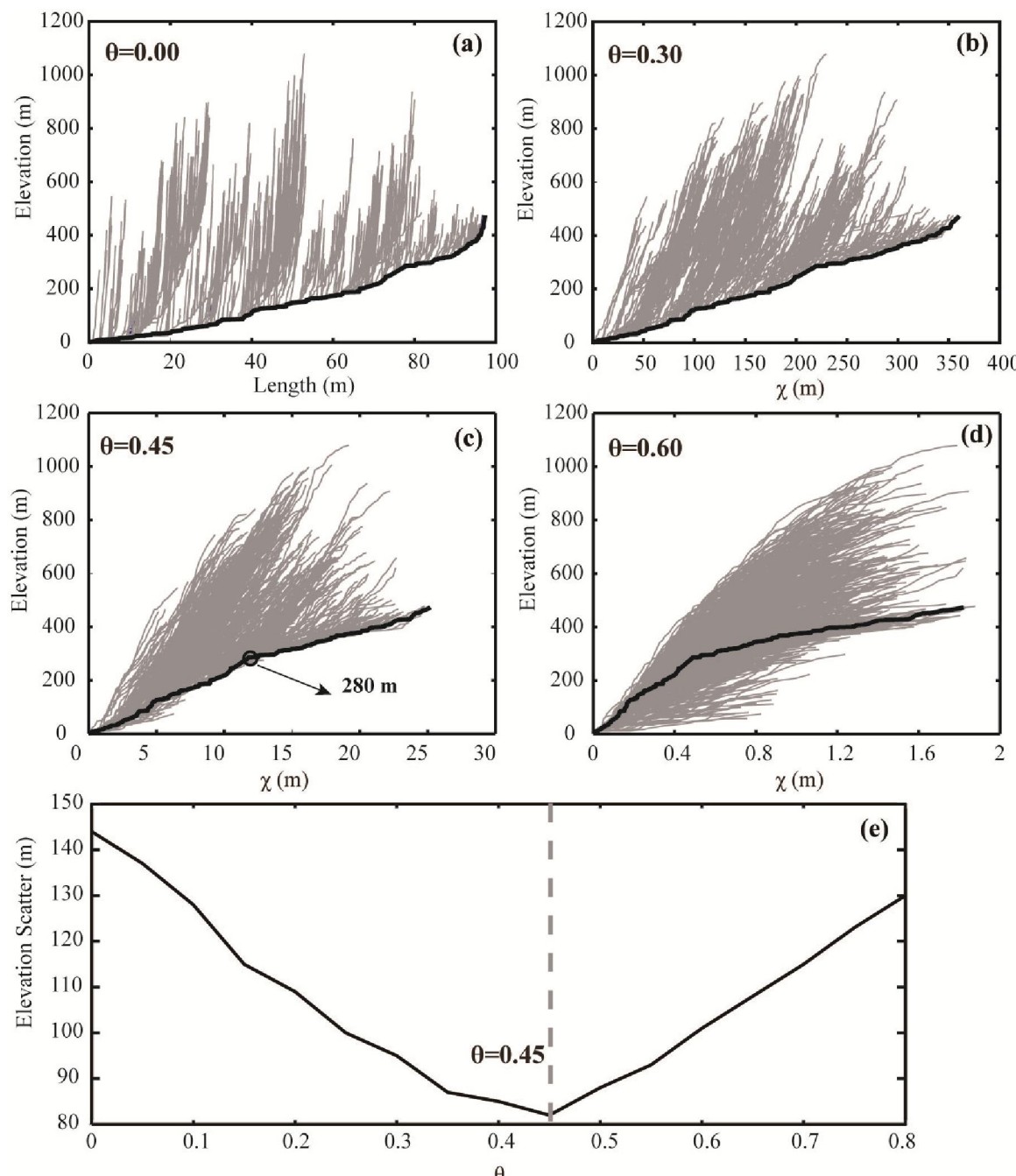

Figure 18.Concavity values that maximize the co-linearity of the main stem with its tributaries.(a)-(d) The χ-z plots of the stem (black line) and its tributaries (gray lines) using different values of $\theta$ ($A_{cr}$=0.1 km$^2$, and $A_0$=1 m$^2$). (e) The elevation scatter of the χ-z plots showing that minimum scatter is achieved with $\theta$=0.45.

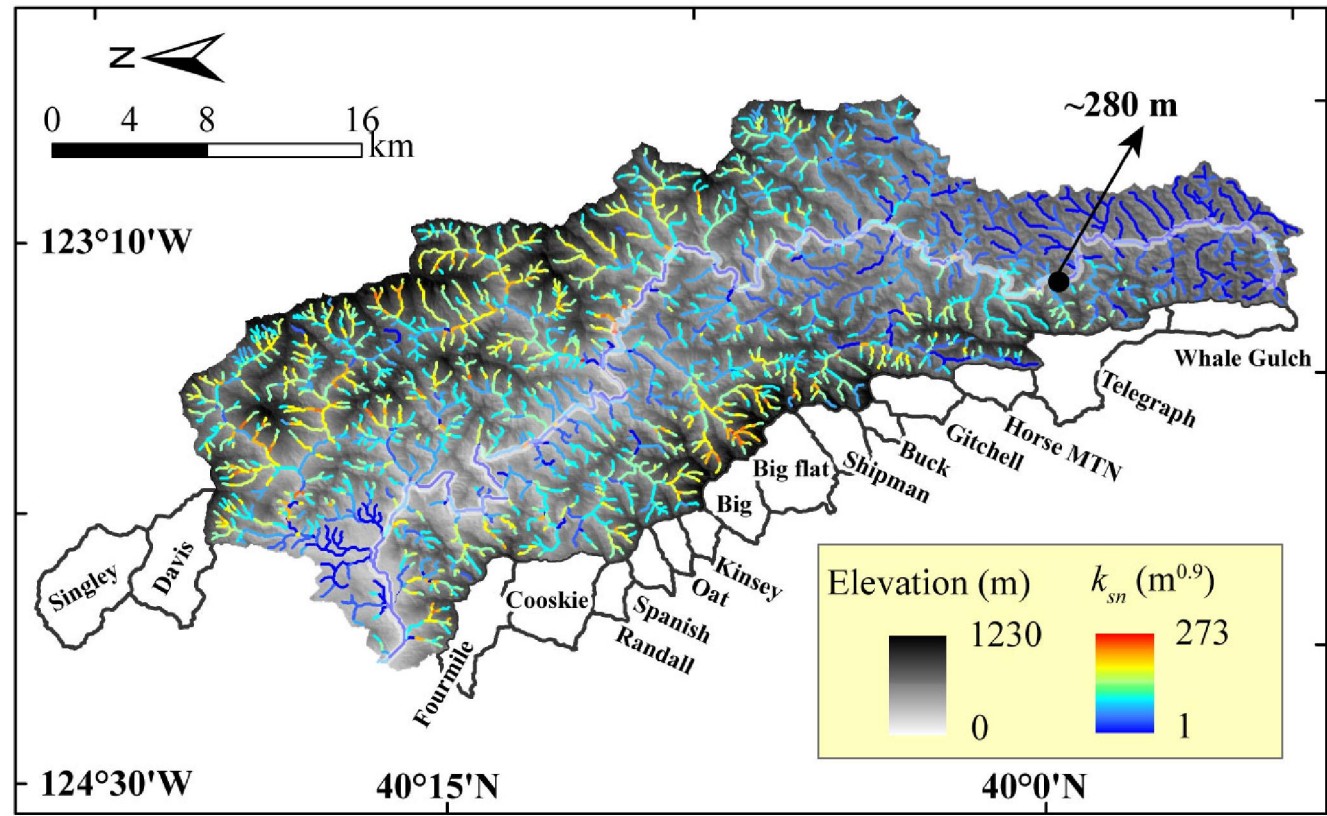

Figure 19. The map of $k_{sn}$ ($\theta$=0.45, elevation interval of 100 m) of the Mattole drainage basin. The black circle indicates the knickpoint on the stem. Low values are shown along the whole stem and its tributaries above the knickpoint. High $k_{sn}$ values are distributed along the upstream of the tributaries below the knickpoint.