# Peer review of "Coupling slope-area analysis, integral approach and statistic tests to steady state bedrock river profile analysis"

_Earth Surface Dynamics, 2016_

## Referee Comment (RC1) · Anonymous Referee #1 · 27 Sep 2016

Wang et al. combine elements of the well-known slope-area and integral approaches to solve the stream power model assuming steady state. First, they estimate a value for concavity index (m/n) using an integral approach. They then determine steepness index. They argue, based on previous work, that slope-area analysis can be used to identify substrate along a river (e.g. bedrock, alluvium). They discuss problems with slope-area analysis including differentiation of discrete and noisy data, which produces unstable results. Solving the integral problem avoids such issues. Statistical tests are performed to 'examine if residuals are independent and homoscedastic'.

I have three comments. My first comment concerns the assumption of steady state. The authors do not provide evidence that rivers analyzed are at steady state. They

refer to Snyder et al. (2000), who used Merritts and Bull's (1989) eustatic correlations to determine uplift rates. Merritts and Bull's calculated uplift rates are very variable in space and time (0-4 mm/yr). Snyder et al. outline their reasons for assuming steady state (e.g. low and constant uplift rates for >100 ka south of the Jackass catchment, stable climate, profile shape; see their pg. 1254). They state that disequilibrium conditions are more likely in regions of high uplift-rate (e.g. the rivers north of ∼40oN in this study). Snyder et al. are circumspect in their assumption of steady state. River shape is not diagnostic of equilibrium conditions. In other places, recent work on inversion of drainage patterns for uplift rate histories indicates that river profile shapes are controlled by spatio-temporal variations in uplift rate moderated by erosional processes (e.g. Pritchard et al., 2009; Roberts & White, 2010; Roberts et al., 2012). The inverse integral approach first described in these papers does not require a priori assumption of steady state.

The assumption of steady state makes it difficult to interpret changes in concavity index reported by the authors. Perhaps the shapes of these rivers are a function of smoothly varying uplift rates and a simpler erosional history? It would be straightforward to test this idea by inverting for uplift rate histories and comparison of results to independent observations of uplift along the Californian coastline.

Secondly, there are a number of relevant papers that are not cited. For example, the integral methodology was developed in a number of papers not discussed (e.g. Pritchard et al., 2009; Roberts & White, 2010; Roberts et al., 2012; Czarnota et al., 2014; Paul et al., 2014; Wilson et al., 2014). Their results suggest that uplift can be inserted along rivers, which makes values of chi difficult to interpret. Erosional parameter values in the stream power model (e.g. m and n) can be determined from joint inversion of drainage patterns (e.g. Rudge et al., 2015). In Roberts et al. (2012, doi:10.1029/2012TC003107) the slope-area methodology was shown to produce unstable results for small amounts of randomly distributed noise.

Finally, the methodology (e.g. lines 26-30 on page 4 and lines 1-4 on page 5) is diffi-
cult to follow. I think the authors suggest that concavity indices vary along rivers with different substrates (e.g. alluvium, bedrock)? And the point of doing the slope-area analysis is to identify where substrate changes? Their approach needs to be explained more clearly. For example, I think caption to Figure 4c could be clearer (e.g. 'The correlation coefficients of chi-z plots as a function of theta for the bedrock portion of the river'). A comparison between predictions of substrate from slope-area analysis and observations would give the reader more confidence in results (e.g. page 4, lines 14 and 26-30). Can substrate be verified using, for example, satellite imagery/the Snyder paper? Some terminology used is confusing. For example, what does 'proper bedrock channel concavity' mean?

Typographical errors. Page 3, line 7: '. . .a better way to [perform] stream profile analysis'. Page 3, line 15: define zb. Page 4, lines 15, 27, 28: add spaces between numbers and units. Caption to Figure 4: labels for panels e and f are incorrect. Figure 4, panel (f): 'aluvial'.

---

## Referee Comment (RC2) · Anonymous Referee #2 · 30 Sep 2016

This contribution compares and combines two approaches for analyzing steady state bedrock channels, the slope-area and integral methods. It uses a well-studied set of streams in tectonically active northern California as its test case. The main finding is that the integral approach yields better-constrained values of channel concavity and steepness parameters. The slope-area method is useful to identify scaling breaks, which are then used as limits over which to apply the integral method. Given the popularity of longitudinal profile analysis in tectonic geomorphology, this kind of methods comparison has value. I have several suggestions for improving the manuscript.

1. The MTJ site is well studied, but has the disadvantage of including only short (<10 km long; <20 km2 area) streams. Choosing a site with longer, larger rivers (and there-

fore more robust scaling between slope or chi and area) would have some merit, given the statistical focus of the study. Longer streams would likely also show clearer downstream transitions between colluvial, bedrock and alluvial conditions. Some of the limitations of the MTJ study area for slope-area analysis where discussed by Wobus et al. (2006), but this study does not make much reference to those findings.

2. The authors use 1 arc-second SRTM DEM, which has a very coarse resolution compared with the 1/3 arc-second DEM available from the USGS (Wobus et al., 2006). Why was the coarse dataset used? Also, how were the profile smoothing parameters chosen (e.g., Figure 3 caption)?

3. In the slope-area method of the uncertainty in steepness partially comes from the uncertainty in concavity. Different concavities result in very different steepnesses; this is the reason that researchers use a reference concavity when comparing channels. The authors do not present an uncertainty on the concavity values found via the integral approach; this may explain the extremely low uncertainties on the steepness values.

4. As others have done in past studies, the authors choose the part of the stream in which to conduct the analysis (the bedrock channels) based on scaling breaks in the slope-area data (Figures 3a and 4a). In the ideal case, these transitions would be mapped in the field, although in practice they are likely gradual and difficult to identify on the ground. Why do the authors believe that the scaling breaks identified in slope-area data are more geomorphically meaningful than those seen in the integral analysis (e.g., Figure 4 analysis; p. 4-5)? I suspect that the authors are assigning too much geomorphic meaning to fairly subtle scaling breaks seen in the slope-area data.

---

## Author Comment (AC1) · 28 Oct 2016

In this contribution, we compare and combine two well-known methods, the slope-area and integral approach, to analyze steady state bedrock channels. Using a well-studied set of streams in Mendocino Triple Junction (MTJ) region of northern California as a test case, we illustrate that The slope-area method is useful to identify scaling breaks (variation in substrate along a river), which are then used as limits over which to apply the integral method. The integral approach yields better-constrained values of channel concavity and steepness parameters with low uncertainty. Statistical tests (Durbin-Watson test, Spearman rank correlation coefficient test) are then performed to examine if residuals are independent and homoscedastic. The coupled process

thus can produce more reliable and robust estimates of uncertainties in stream power parameters.

---

## Author Response (AR1)

Wang et al. combine elements of the well-known slope-area and integral approaches to solve the stream power model assuming steady state. First, they estimate a value for concavity index (m/n) using an integral approach. They then determine steepness index. They argue, based on previous work, that slope-area analysis can be used to identify substrate along a river (e.g. bedrock, alluvium). They discuss problems with slope-area analysis including differentiation of discrete and noisy data,

10 which produces unstable results. Solving the integral problem avoids such issues. Statistical tests are performed to 'examine if residuals are independent and homoscedastic'.

I have three comments. My first comment concerns the assumption of steady state. The authors do not provide evidence that rivers analyzed are at steady state. They refer to Snyder et al. (2000), who used Merritts and Bull's (1989) eustatic

15 correlations to determine uplift rates. Merritts and Bull's calculated uplift rates are very variable in space and time (0-4 mm/yr). Snyder et al. outline their reasons for assuming steady state (e.g. low and constant uplift rates for >100 ka south of the Jackass catchment, stable climate, profile shape; see their pg. 1254). They state that disequilibrium conditions are more likely in regions of high uplift-rate (e.g. the rivers north of 40oN in this study). Snyder et al. are circumspect in their assumption of steady state. River shape is not diagnostic of equilibrium conditions. In other places, recent work on inversion

20 of drainage patterns for uplift rate histories indicates that river profile shapes are controlled by spatio-temporal variations in uplift rate moderated by erosional processes (e.g. Pritchard et al., 2009; Roberts & White, 2010; Roberts et al., 2012). The inverse integral approach first described in these papers does not require a priori assumption of steady state.

The assumption of steady state makes it difficult to interpret changes in concavity index reported by the authors. Perhaps the shapes of these rivers are a function of smoothly varying uplift rates and a simpler erosional history? It would be

25 straightforward to test this idea by inverting for uplift rate histories and comparison of results to independent observations of uplift along the Californian coastline.

Thanks for the comment. River shape is not diagnostic of equilibrium conditions despite linear fit to the log-transformed slope-area data or the χ-*z* plot.

30 In the low-uplift zone (streams Hardy to Dehaven), the uplift rate has been approximately constant (rather than smoothly varying values) for at least 0.33 Ma (Merritts and Bull, 1989). The bedrock-channel reaches upstream are probably not affected by sea-level fluctuations (Snyder et al., 2000). These streams thus can be under steady state. Higher concavity values in the lower reaches may be due to sedimentation affected by sea-level fluctuations.

However, disequilibrium conditions are likely in the high-uplift zone. To test the steady assumption, we modeled uplift rate histories (see Sect. 4.2 for details).

For spatially variant rock uplift, the study area can be divided into four distinct zones, from north to south, the north transition zone (streams Singley to Cooskie), the King Range high-uplift zone (streams Randall to Buck), the intermediate-uplift zone (streams Horse Mtn), and the low-uplift zone (streams Hardy to Dehaven). Within each zone, we assumed spatially invariant rock uplift for its small drainage area. We extracted all the fluvial channels and calculated a mean $\chi$-$z$ plot for each zone. Base on $n$=1 and variable erodibility $K$ (relates to uplift rates), we utilized a linear inversion model of Goren et al (2014) to inferred the rock uplift histories $U(t)$.

As shown by Fig. 15j, the rock uplift rates in the low- and intermediate-uplift zones have been constant (~0.3-0.4 mm/a since 0.4 Ma, and ~2-2.5 mm/a since 0.16 Ma, respectively). The north transition and high-uplift zones both experienced increases in uplift rates (from ~2.5 mm/a to 3.3 mm/a, and from ~3.7 mm/a to 4.3 mm/a, respectively) starting ~0.12 Ma ago. However, the increase ratios are much lower. The uplift rates have been constant for at least 0.12 Ma (the maximum response time is ~0.16 Ma) and no knickpoints are found along the rivers, which means that the river channels have been reshaped by the recent tectonic activity and already have reached steady state.

Secondly, there are a number of relevant papers that are not cited. For example, the integral methodology was developed in a number of papers not discussed (e.g. Pritchard et al., 2009; Roberts & White, 2010; Roberts et al., 2012; Czarnota et al., 2014; Paul et al., 2014; Wilson et al., 2014). Their results suggest that uplift can be inserted along rivers, which makes values of chi difficult to interpret. Erosional parameter values in the stream power model (e.g. m and n) can be determined from joint inversion of drainage patterns (e.g. Rudge et al., 2015). In Roberts et al. (2012, doi:10.1029/2012TC003107) the slope-area methodology was shown to produce unstable results for small amounts of randomly distributed noise.

Thanks for the comment and we have cited these papers. We cite papers (Pritchard et al., 2009; Roberts and White, 2010; Roberts et al., 2012; Czarnota et al., 2014; Paul et al., 2014; Wilson et al., 2014) in Line33, Page9.

We cite the paper (Rudge et al., 2015) in Line19 Page7.

We cite the paper (Roberts et al., 2012) in Line22 Page8. Roberts et al (2012) noticed that the slope-area methodology may produce unstable results for small amounts of randomly distributed noise. In spite of little knowledge about the elevation data uncertainty here, we utilized different datasets and various data handling methods (data smoothing and sampling) to calculate slope with different uncertainties. Then, to some extent, the influence of data uncertainty can be tested (see Sect. 4.3 for details).

In the case study, the channel slope is derived from 1 Arc-second SRTM DEM via 300 m smoothing window and 20 m contour sampling interval. Then, we reanalyzed streams in the high- and low-uplift zones based on 1/3 arc-second USGS DEM. We calculated the channel slope via various strategies: 300 m smoothing window and 20 m contour sampling interval, 300 m smoothing window and 10 m contour sampling interval, and 100 m smoothing window and 10 m contour sampling

interval, respectively. We found no distinct difference is in concavity and channel steepness values derived from different datasets or data handling methods. Therefore, in this study area, uncertainty in elevation data may not cause distinct differences in parameter estimates (e.g. $\theta$ and $k_{sn}$).

5   Finally, the methodology (e.g. lines 26-30 on page 4 and lines 1-4 on page 5) is difficult to follow. I think the authors suggest that concavity indices vary along rivers with different substrates (e.g. alluvium, bedrock)? And the point of doing the slope-area analysis is to identify where substrate changes? Their approach needs to be explained more clearly. For example, I think caption to Figure 4c could be clearer (e.g. 'The correlation coefficients of chi-z plots as a function of theta for the bedrock portion of the river'). A comparison between predictions of substrate from slope-area analysis and observations

10   would give the reader more confidence in results (e.g. page 4, lines 14 and 26-30). Can substrate be verified using, for example, satellite imagery/the Snyder paper? Some terminology used is confusing. For example, what does 'proper bedrock channel concavity' mean?

Thanks for the comment. We suggest that concavity indices vary along rivers with different substrates (e.g. alluvium,

15   bedrock). The point of doing the slope-area analysis is to identify where substrate changes. We have revised the text (lines 1-19 on page 5) and the caption to Figure 4c (The correlation coefficients of chi-z plots as a function of $\theta$ for the bedrock portion of the river).
    The substrate of the streams in the case study has been verified by Snyder et al (2000, 2003).
    We have rectified the confusing terminology 'proper bedrock channel concavity' as 'concavity of the bedrock portion of

20   the river' in Line5 Page5.

Typographical errors. Page 3, line 7: '…a better way to [perform] stream profile analysis'. Page 3, line 15: define zb. Page 4, lines 15, 27, 28: add spaces between numbers and units. Caption to Figure 4: labels for panels e and f are incorrect. Figure 4, panel (f): 'aluvial'.

25   Thanks for the comment and we have rectified these errors.
    The coupled process does provide a better way to perform stream profile analysis (Line10, Page3).
    We define $z_b$ as the channel elevation at $x$=0 (river mouth). Line 7 page 2.
    We have added spaces between numbers and units Page 4, lines 19; Page5 Lines 2, 3.
    We have rectified the caption to Fig. 4 and the word 'alluvial' in Fig. 4f.
This contribution compares and combines two approaches for analyzing steady state bedrock channels, the slope-area and integral methods. It uses a well-studied set of streams in tectonically active northern California as its test case. The main finding is that the integral approach yields better-constrained values of channel concavity and steepness parameters. The slope-area method is useful to identify scaling breaks, which are then used as limits over which to apply the integral method.

10 Given the popularity of longitudinal profile analysis in tectonic geomorphology, this kind of methods comparison has value. I have several suggestions for improving the manuscript.

1. The MTJ site is well studied, but has the disadvantage of including only short (<10 km long; <20 km2 area) streams. Choosing a site with longer, larger rivers (and therefore more robust scaling between slope or chi and area) would have some

15 merit, given the statistical focus of the study. Longer streams would likely also show clearer downstream transitions between colluvial, bedrock and alluvial conditions. Some of the limitations of the MTJ study area for slope-area analysis where discussed by Wobus et al. (2006), but this study does not make much reference to those findings.

Thanks for the comment. The case study has disadvantages of including only short (<10 km long; <20 $km^2$ area) and

20 steady streams. In many landscapes, especially large rivers, this steady assumption will not be met. Then, we took Mattole, a large river in the MTJ region, for example, to extract the unsteady signals. Here, 1/3 arc-second USGS DEM was used (see Sect. 4.4 for details).

We derived a log-transformed slope-area plot of the stem (300 m smoothing window and 20 m contour sampling interval) and recognized a knickpoint by the scaling break in the slope-area data. The knickpoint locates at the elevation of ~280 m.

25 Concavity indices above (0.61±0.01) and below (0.58±0.07) the knickpoint are nearly the same.

Using the integral approach and two statistic tests, we derived the $k_{sn}$ above (10.81±0.86 $m^{0.9}$) and below (17.44±1.16 $m^{0.9}$) the knickpoint. The $k_{sn}$ values of the Mattole stem are much lower than that of the adjacent streams (e.g. Singley, Davis, Fourmile and Cooskie), suggesting that other variables (e.g. sediment flux and lithology) may affect channel steepness. This might limit our ability to quantitatively relate steepness indices to uplift rates in this field setting, as noticed by Wobus et al

30 (2006).

Usually, the method of best linearizing χ-*z* plot can be used to compute *θ* of a steady-state bedrock channel. However, in many cases, whether a stream is under steady state is unknown. We computed the correlation coefficients between channel elevation and χ values of the stem based on a range of *θ*. Different from the result of slope-area analysis, the best linear fit

corresponds to $\theta$=0.30. Thus, a river may be in disequilibrium condition despite a linear relationship in the $\chi$-z plot. Therefore, in this case, slope-area analysis might be a good choice to determine whether the river profile is under steady state, although the difference in channel substrates is unrevealed from the log-transformed slope-area data.

We calculated the map of channel steepness. The channels show low $k_{sn}$ values along the whole stem and its tributaries (low elevation) above the knickpoint but higher values in the upstream (high elevation) of tributaries below the knickpoint. Among the tributaries in the west of the stem, channel steepness decreases from the central part (high-uplift zone) towards both north (north transition zone) and south (intermediate-uplift zone). Both the spatial pattern of $k_{sn}$ and positive relationship between $k_{sn}$ and elevation may indicate a tectonic control on channel steepness despite other potential variables.

2. The authors use 1 arc-second SRTM DEM, which has a very coarse resolution compared with the 1/3 arc-second DEM available from the USGS (Wobus et al., 2006). Why was the coarse dataset used? Also, how were the profile smoothing parameters chosen (e.g., Figure 3 caption)?

Thanks for the comment. In Figs. 3, 4 and 5, a method of 300 m smoothing window and 20 m contour sampling interval was chosen.

Roberts et al (2012) noticed that the slope-area methodology may produce unstable results for small amounts of randomly distributed noise. In spite of little knowledge about the elevation data uncertainty here, we want to utilize different datasets (1/3 arc-second USGS DEM and 1 arc-second SRTM DEM) and various data handling methods (smoothing window and contour sampling interval) to calculate channel slope with different uncertainties. Then, to some extent, the influence of data uncertainty can be tested (see Sect. 4.3 for details).

3. In the slope-area method of the uncertainty in steepness partially comes from the uncertainty in concavity. Different concavities result in very different steepnesses; this is the reason that researchers use a reference concavity when comparing channels. The authors do not present an uncertainty on the concavity values found via the integral approach; this may explain the extremely low uncertainties on the steepness values.

Thanks for the comment. For the integral approach, two methods are used to determine the channel concavity: 1) best linearize the $\chi$-z plot of the stem; 2) maximizes the co-linearity of the main stem with its tributaries. Based on either method, we can only derive a unique concavity value (It is different with slope-area analysis).

For steady streams with uniform rock uplift rates, lithology and climate, the two kinds of $\theta$ values can be similar. The limited difference between the two values can be an uncertainty estimate of concavity. In this study, we utilized both methods to calculate the channel concavity values to see whether they are similar (e.g. Lines16-21, Page 6).

For the same stream, distinct difference in the two kinds of concavity values may suggest other potential variables, for example, unsteady channels (Lines 19-20, 30 Page 9) and variant substrates (Lines22-30, Page 6).

4. As others have done in past studies, the authors choose the part of the stream in which to conduct the analysis (the bedrock channels) based on scaling breaks in the slope-area data (Figures 3a and 4a). In the ideal case, these transitions would be mapped in the field, although in practice they are likely gradual and difficult to identify on the ground. Why do the authors believe that the scaling breaks identified in slope area data are more geomorphically meaningful than those seen in the integral analysis (e.g., Figure 4 analysis; p. 4-5)? I suspect that the authors are assigning too much geomorphic meaning to fairly subtle scaling breaks seen in the slope-area data.

Thanks for the comment. The transitions between colluvial, bedrock and alluvial portions of the channel have been verified by Snyder et al (2000, 2003). In Fig. 4, we show that concavity indices vary along rivers with different substrates (e.g. alluvium, bedrock). The variation in $\theta$ can be seen directly in the slope-area plot.

We do not mean that the scaling breaks identified in slope-area data are more geomorphically meaningful than those seen in the integral analysis. In fact, they are with different geomorphically meanings.

In Fig.4, the scaling break identified in slope-area data shows difference in channel concavity values. Usually, channel concavity is more related to lithology than channel steepness.

The scaling break in $\chi$-$z$ plot shows difference in channel steepness rather than concavity indices. This is useful to extract the rock uplift history (In Sect. 4.2, we modeled the uplift histories based on the $\chi$-$z$ plots of the streams).

[revised manuscript text omitted]

---

## Author Response (AR2)

**Associate Editor Decision: Publish subject to minor revisions (review by Editor)** (05 Jan 2017) by Prof Simon Mudd

Comments to the Author:

Review of Wang et al.

This paper contains a clear comparison of the chi and S-A methods for extracting information such as the channel steepness

5   and concavity index from river profiles. The paper is timely since these two methods are being used in parallel and a comparison of their strengths yet to be performed in such detail. I ask for revisions as I am not quite prepared for this to be published; I believe there are some elements that could be improved and will hopefully increase the impact of the paper. The original AE was unable to continue handling the paper, and thus I have looked at this paper after the first round of revisions, so this does constitute a second round of review that the authors were not expecting. Hopefully the authors find my

10   comments constructive. I have spent some time making editorial comments in order to improve the readability of the manuscript. In addition, I still feel the authors have missed a few relevant papers, although they have done a good job of addressing the comments relating to landscape transience and the papers cited by one of the referees. Referee 1 had three main comments. Firstly, they felt changes in concavity might have been caused by transience and suggested an inversion of the uplift rates from the channel profiles. The authors have done this to show there is only minimal

15   evidence for transience in the studied catchments, and in addition cite previous work that suggests a low degree of transience. In chi space, the best fit concavity index, based on collinearity, tends to lead to concave channels (e.g., figs 10, 12 and 14). If the channels were all incising into spatially homogenous bedrock, and the uplift rate was spatially homogenous, this would suggest a slowing of uplift. This is seen in the inversion results, but is subtle. It is not really clear how the inversion might deal with changing transport processes: the authors suggest the change in concavity is due to alluviation rather than changing

20   uplift rates, and thus represents a change from detachment to transport limited conditions. Overall, I feel the first two comments from this referee are addressed but I do feel the methodology could be clearer. In my lined comments below I make specific suggestions about where more detail is needed.

Referee #2 had further methodological questions. The authors have answered some of the comments but again I feel there is more scope for making their methods clearer. Again, I will suggest changes in my lined comments below.

25   My recommendation is that the paper be accepted subject to revisions. I think there are some interesting and timely results here, but the paper could use some further edits to help readers understand what the authors have done. Since I am not calling for additional analysis, I will say these required revisions are minor.

Lined comments:

30   P1 line 8: Delete "one" after "former". Also what are "deviated concavities"? I don't see a discussion of this in the paper. It seems the concavity indices identified by the S-A method are similar to the ones determined by the chi method, but with higher uncertainties. I would just delete the phrase "and deviated concavities" from this sentence.

Thanks for the comment. We have deleted "one" and "and deviated concavities" and revised the sentence in Line 7-9 P1.

P1. Line 9: "False knickpoints": again, I don't see much discussion of this in the text. How is one more likely to find a "false knickpoint" using chi analysis vs S-A analysis? S-A analysis, done with intensive smoothing, might only pick up the largest knickpoints, but this isn't discussed in the text. The main focus of the text is that the S-A method is better at identifying changes in concavity than the chi method. So why not write that in the abstract instead?

Thanks for the comment. We have revised the sentence as "The former is better at identifying changes in concavity indices but produces stream power parameters with higher uncertainties than the integral approach. The latter is much better for calculating channel steepness." in Line 7-9 P1.

P1. Line 11: Delete "manage to".

Thanks for the comment. We have deleted "manage to" in Line 12 P1.

P1, line 13: Add a sentence explaining why serial correlation of residuals is important to eliminate.

Thanks for the comment. We have add a sentence "In addition, we run bi-variant linear regression statistic tests for the two methods to examine and eliminate serially correlated residuals because they may bias both the estimated value and precision of stream power parameters." to explain the importance of eliminating serial correlation of residuals in Line 12-14 P1.

P2, line 10: This section is not as clear as it should be. There is a very important point that belongs here and that the authors imply but do not state explicitly. The concavity index can change. In the S-A approach, the plots are simply data reflecting topography and the concavity index can be extracted from regression. This regression is subject to the noise of S-A data and the details of profile smoothing, but it is a reflection of data. The chi transformation, on the other hand, contains an assumption about concavity. Thus if a chi profile is used one must use some discriminatory technique to separate areas of different concavity: such a method has not, to my knowledge, been suggested. There are a number of suggestions for determining concavity of a river network using chi analysis but none for determining spatially varying concavities. In S-A analysis the concavity can be extracted directly from a regression of the data. This point is extremely important! I believe this point is what the authors wanted to highlight in the paper so it needs to be stated very, very clearly. S-A analysis is very noisy compared to chi analysis, so this property, that S-A analysis is a more direct measure of concavity, is really the main reason to continue using S-A analysis. The authors would do well to emphasize this point.

Thanks for the comment. We have revised the text as "Concavity changes with different channel substrate properties, which can be reflected and extracted from the slope-area data directly. Then, one can discriminate channel properties according to variable concavity indices." (Line 10-11 P2) and "Nevertheless, the $\chi$ transformation contains an assumption of single concavity, which is distinctly different from slope-area analysis. In fact, concavity can change. In places where there is spatially varying concavities (because channels may go from bedrock to alluvial), the integral approach may show a break in the $\chi$-$z$ plot. Methods of separating areas of different concavities from a $\chi$-$z$ plot have not been suggested. Despite a very

noisy method compared to the integral approach, slope-area analysis is a more direct measure of concavity, which is the main reason to continue using it." (Line 24-28 P2).

P2, line 10: I do suggest rewriting this section, as outlined above, but "for its need to estimate slope" is awkward and needs
5 to be rewritten.

Thanks for the comment. We have revised the text as outlined above (Line10-11 and Line24-28, P2). We have rewritten the sentence as "However, estimates of slope obtained by differentiating and resampling noisy elevation data are even noisier." In Line 13-14 P2.

10 P2 line 12-13: Awkward sentence. Rewrite to: "In addition, the derived channel steepness suffers from high uncertainty due to…". Also Perron and Royden need to be re-cited here.

Thanks for the comment. We have rewritten the sentence as "In addition, the derived channel steepness suffers from high uncertainty due to error propagation (Perron and Royden, 2013)" in Line 16 P2.

15 P2 lines 19-21: This section needs some rewriting. Say "Thus, the chi method provides…". Then the reason the uncertainty will be underestimated should be explained. Finally the last sentence ("false concavities and knickpoints") needs to be rewritten into my suggestions stated above: because chi analysis is based on the assumption of a single concavity, in places where there is spatially varying concavities (because channels may go from bedrock to alluvial) the chi method may show a break in the chi-elevation profile that is spurious.

20 Thanks for the comment. We have revised the text as "the $\chi$ transformation contains an assumption of single concavity, which is distinctly different from slope-area analysis. In fact, concavity can change. In places where there is spatially varying concavities (because channels may go from bedrock to alluvial), the integral approach may show a break in the $\chi$-$z$ plot. Methods of separating areas of different concavities from a $\chi$-$z$ plot have not been suggested. Despite a very noisy method compared to the integral approach, slope-area analysis is a more direct measure of concavity, which is the main reason to
25 continue using it." in Line 24-28 P2.

We have explained the reason the uncertainty will be underestimated as "In addition, the uncertainty in $k_s$ will be underestimated because the transformed profile ($\chi$-$z$ plot) is a continuous curve, and therefore the residuals of the linear fit are serially correlated (Perron and Royden, 2013)." in Line 28-30 P2.

30 P3 Line 1: say "is debris flow dominated and therefore will not display the typical fluvial scaling in Equaiton (1)" and then cite the Stock and Dietrich 2003 paper:

Stock, J., and W. E. Dietrich (2003), Valley incision by debris flows: Evidence of a topographic signature, Water Resour. Res., 39(4), 1089, doi:10.1029/2001WR001057.

Thanks for the comment. We have revised the text as "is debris flow dominated and therefore will not display the typical fluvial scaling in Eq. (1) (Stock and Dietrich, 2003)" in Line 10 P3.

P3 Line 3: the sentence ending in "scaling" needs a citation. The Whipple and Tucker 1999 paper might be appropriate here, or perhaps the 1994 willgoose paper:

Willgoose, G. (1994), A physical explanation for an observed area-slope-elevation relationship for catchments with declining relief, Water Resour. Res., 30(2), 151–159, doi:10.1029/93WR01810.

Thanks for the comment. We have cited the papers (Whipple and Tucker, 1999; Willgoose, 1994) in Line 12 P3.

P3, line 5. Replace "could get" with "derive".

Thanks for the comment. We have replaced "could get" with "derive" in Line 15 P3.

P3, line 7: This sentence is unclear. How exactly is the channel subject to lower uncertainties? You can restate that the chi analysis of the channel can be derived from the bedrock section that is suggested by the varying concavities in S-A space.

Thanks for the comment. We have revised the text as "the $\chi$-$z$ analysis of the channel can be derived from the bedrock section that is suggested by the varying concavities in slope-area space" in Line16-17 P3.

P3, Line 9: Here the authors say that the coupled process provides a better way to perform stream profile analysis. As I mention earlier, this has not been well demonstrated because the different advantages of the two methods have not been sufficiently highlighted in previous sections. Modification of the text earlier in the paper should fix this problem.

Thanks for the comment. We have sufficiently highlighted different advantages of the two methods in previous sections (Line 9-30 P2).

Page 3, Line 9: While it may be true that "little has been done" on the statistics of S-A or the integral method, there are some papers that investigate this topic that are strangely not cited here. In fact there is a paper with the title "A statistical framework to quantify spatial variation in channel gradients using the integral method of channel profile analysis" which I am familiar with because I wrote it! The entire purpose of that paper was to explore statistical ways to constrain channel steepness and m/n. So I cannot understand why that paper is not even mentioned.

Mudd, S. M., M. Attal, D. T. Milodowski, S. W. D. Grieve, and D. A. Valters (2014), A statistical framework to quantify spatial variation in channel gradients using the integral method of channel profile analysis, J. Geophys. Res. Earth Surf., 119(2), 2013JF002981, doi:10.1002/2013JF002981.

As a historical note, our original version of that paper included calculation of the D-W statistic, but we found it did not yield information that was better than other tests. However, if you use our code that was released along with the 2014 paper you

will see that it still spits out the Durbin-Watson statistics for all chi analysis (https://csdms.colorado.edu/wiki/Model:Chi_analysis_tools).

Thanks for the comment. We are sorry for missing the paper. We have added the paper as "Mudd et al (2014) proposed a statistical framework to quantify spatial variation in channel gradients and calculated Durbin-Watson statistics in their code (https://csdms.colorado.edu/wiki/Model:Chi_analysis_tools)." in Line 24-25 P3.

P3, line 14: What I think is missing from this section is a clear description of why these statistical tests are necessary. I only think a few sentences are necessary but an explanation of the benefit of these statistics would help keep the attention of the reader.

Thanks for the comment. We have explained why these statistical tests are necessary in Line 20-27 P3. The text is as below:
The coupled process does provide a better way to perform stream profile analysis. Indeed, both the slope-area analysis and integral approach are bi-variant linear regression methods. Statistically, some tests must be done to meet two critical conditions, i.e. the residuals are independent and homoscedastic (Cantrell, 2008; Kirchner, 2001). Perron and Royden (2013) noticed that the precision in steepness derived from the integral approach would be overestimated due to auto-correlated residuals. Mudd et al (2014) proposed a statistical framework to quantify spatial variation in channel gradients and calculated Durbin-Watson statistics in their code (https://csdms.colorado.edu/wiki/Model:Chi_analysis_tools). In this contribution, we find that auto-correlation of residuals can bias the regression coefficient, channel steepness (see details in Sect. 3). Therefore, not only theoretically but in practice, statistical tests are necessary for both the two methods.

P.3, line 16: The authors need to state if these tests are performed on the entire profile or only for the sections identified as bedrock using the S-A analysis.

Thanks for the comment. These tests are performed on the sections identified as bedrock using the slope-area analysis. (Line 29-30 P3)

P4, line 11: Again, an additional sentence or two is needed here to explain why these statistics are needed and useful.

Thanks for the comment. We have explained it in Line 20-27 P3.

P4, line 17: The Harkins et al is an excellent paper, but they do not present any evidence that chi and S-A methods produce the same results. To my knowledge, the first paper to actually compare the methods was the paper by Scherler et al (2014), which should be cited here:

Scherler, D., B. Bookhagen, and M. R. Strecker (2014), Tectonic control on 10Be-derived erosion rates in the Garhwal Himalaya, India, J. Geophys. Res. Earth Surf., 119(2), 2013JF002955, doi:10.1002/2013JF002955.

Thanks for the comment. We have cited the paper in Line 6 P5.

P4, line 19: How was the dividing point determined? Was it done by eye or some other method? This needs to be explained. There are a number of ways to do this rigorously. I've done it with multiple segments in the Mudd et al. 2014 JGR-ES paper and we also used a 2 segment method for 1st order channels to find the area of the process transition in this paper:

5    Clubb, F. J., S. M. Mudd, D. T. Milodowski, M. D. Hurst, and L. J. Slater (2014), Objective extraction of channel heads from high-resolution topographic data, Water Resour. Res., 50(5), 4283–4304, doi:10.1002/2013WR015167.

Thanks for the comment. We have explained it in Line 7-11 P5. The text is as below:

The area of process transition along a river profile can be determined by some rigorous methods. For example, Mudd et al (2014) worked on it with multiple segments and Clubb et al (2014) used a two-segment method for 1st order channels to find

10   the area of process transition. Nevertheless, Fig. 3a shows a very simple log-transformed slope-area plot, from which the colluvial (nearly constant log(slope) ~ -1) and fluvial (decreasing channel gradient) sections can be discriminated just by eye.

P4, line 23: The difference in the results between this paper and the Perron and Royden paper is curious. If you use the 10m DEM that they used do you get the same result they did? I note that a previous review has asked for such an analysis and it

15   has not been performed. I now see that higher resolution data has been downloaded and analysed so it seems not too much work to report the results with higher resolution data.

Thanks for the comment. We use the 10m DEM for stream Cooskie and get two kinds of concavity values: 0.49 (slope-area analysis), and 0.51 (the integral approach) (see the figure as is shown below). These results are similar to Fig. 3 in our contribution (1Arc SRTM DEM is used) but different with Perron and Royden (2013). That's because we choose a tributary

20   that is different to Perron.

There are many tributaries in the catchment Cooskie. The length of the tributary used by Perron is ~2.5 km (Fig. 1a in Perron and Royden (2013)), while that we used is longer than 7 km (Fig. 3b in our paper, or Fig b as is shown below). The stream used in our work is mapped in Fig. 1. Nevertheless, we do not know which tributary (in Cooskie catchment) was used in Perron and Royden (2013) because they did not show a map of the tributary they used.

[Figure]

Stream profile analysis of Cooskie. Elevation data are from 1/3 arc-second USGS DEM. (a) Log-transformed slope-area plot. The slope was derived from the smoothed (horizontal distance of 100m) and re-sampled (elevation interval of 10m) elevation data. (b) The full river profile (without any smoothing or re-sampling) of Cooskie. (c) The correlation coefficients, R, as a function of $\theta$ for least-squares regression based on Eq. (5). The maximum value of R, which corresponds to the best linear fit, occurs at $\theta$=0.51 (dotted line and black arrow). (d) $\chi$-$z$ plot of the bedrock channel profile, transformed according to Eq. (3) with $\theta$=0.51, $A_{cr}$=0.1 km$^2$, and $A_0$=1 m$^2$.

P4. Line 24: Delete "with each other". Then later in the sentence you can say the "there are large uncertainties in channel steepness of approximately 40%. Also I think there should be a sentence saying how the uncertainty is computed.

Thanks for the comment. We have deleted "with each other". The uncertainties are calculated by dividing the estimated value by error (see details in Line 17-18 P5). For example, the steepness value is $k_s$=79.16±29.35. Then, the uncertainty is (29.35/79.16) ~ 40%.

P4, line 30: This is an important result and I feel it should be stated in the abstract.

Thanks for the comment. We have stated in the abstract (Line 9 P1).

P5, line 3, 4: Again, how was this transition point identified? The method needs to be stated.

Thanks for the comment. We determine the transition point where channel concavity changes (from the log S-A plot). We have stated it in Line 7-11 and 26-26 P5.

P5, lines 13-15: These sentences could be clearer. Please rewrite.

Thanks for the comment. We have rewritten the sentences as "However, no knickpoint occurs on stream Juan because the river has been controlled by uniform rock uplift and under steady state (Snyder et al., 2000; see Sect. 4.2 for discussion). Thus, a $\chi$-$z$ plot generated by a single concavity will lead to misestimates in stream power parameters. We should recognize changes in concavities from slope-area space." in Line 4-7 P6.

P5, lines 20-24: Again, there is a need for some text explaining why the D-W statistic and t statistics are useful. What are the implications of these tests?

Thanks for the comment. We have explained why the two statistic tests are useful. The texts are as below:

(Line 16-22 P6) The uncertainties in steepness indices (revised by Durbin-Watson test) are about 2.4% - 9.9% (Fig. 8), which are much higher than those without statistical test (lower than 1%, Fig. 6). In addition to uncertainty estimate, auto-correlated residuals can also bias the regression coefficient, steepness. The channel steepness values of streams Fourmile, Kinsey and Hardy are 57.01, 103.90 and 58.78 $m^{0.9}$ (Fig. 6). While revised by Durbin-Watson test, these values are 36.33, 82.16 and 76.65 $m^{0.9}$, respectively (Fig. 8). Then, steepness varies about 25.6% - 58.3% (dividing the difference of the two kinds of steepness indices by the values revised by Durbin-Watson test). Due to such influence of auto-correlated residuals on both the estimated value and precision of steepness, Durbin-Watson test is necessary when applying the integral approach.

(Line 25-26 P6) Despite no heteroscedasticity found in our study area, we suggest that Spearman rank correlation coefficient test should also be done because the test is a part of linear regression (statistically).

p.6, line 3: Here is another chance for the authors to highlight why the S-A approach is useful even though they have shown that it gives highly uncertain channel steepness values. They need to reiterate here that S-A data makes no assumptions about m/n and therefore is more sensitive than chi analysis in detecting spatially varying concavities.

Thanks for the comment. We have revised the text as "Even though it gives highly uncertain channel steepness values, slope-area plot makes no assumptions about $\theta$ and therefore is more sensitive than $\chi$ analysis in detecting spatially varying concavities." in Line 2-3 P7.

P6, line 8-9: Awkward sentence. Rewrite.

Thanks for the comment. We have revised the text as "Perron and Royden (2013) considered that the uncertainty in channel concavity derived from a linear regression of the log-transformed slope–area plot described how precisely one can measure the slope of the plot, not how precisely the parameter is known for a given landscape." in Line 9-11 P7.

P6, line 13: In the Mudd et al., 2014 paper we included an extensive discussion of selecting m/n based on the main stem vs the collinearity and I feel that paper should be cited here.

Thanks for the comment. We have cited the paper in Line14 P7.

P6, line 15: I suggest writing Theta_co is the concavity from the collinearity test, and theta_mR is the concavity derived from averaging. Also, there should be some more detail as to how theta_mR is calculated: what, exactly, is being averaged?

Thanks for the comment. We have revised the text as "$\theta_{Co}$ (derived from the collinearity test) and $\theta_{mR}$ (from averaging the concavity values of all the streams within a catchment)" in line 17-18 P7.

P6, line 30: It needs to be stated here if these results are based on all of the profile data or just the data that has been determined to be bedrock using S-A analysis.

Thanks for the comment. We have stated that these results are "for all of the profile data" in Line 1 P8.

15 Page 8, line 6: Replace "various" with "variable". Do you mean "calculate"? This sentence doesn't make sense, please check.

Thanks for the comment. We have revised the text as "We utilized variable erodibility ($K=U/k_{sn}$) values to calculate rock uplift rates." in Line 8 P9.

20 P.8 Line 22: Add sentence explaining why this is the case.

Thanks for the comment. We have explained it as "Roberts et al (2012) noticed that the slope-area methodology might produce unstable results because small amounts of randomly distributed noise added to river profile will cause significant change in channel gradient." in Line 24-25 P9.

25 P. 8 line 33: Unclear what is meant by "composited". Reword.

Thanks for the comment. We have revised the text as "slope-area data from all the streams within the same zone were composited." in Line 2 P10.

P.9 line 10: This is an interesting result, but brings me back to the question of why your estimated concavity of the bedrock
30 sections is so different from that estimated by Perron and Royden (2013); see comment on p4, line 23.

Thanks for the comment. We have explained it as "We do not deny that utilizing different datasets may cause some difference in parameter estimate for an individual catchment. For example, when using the integral approach, the resulting channel concavity of stream Cooskie is 0.45 (in Sect. 3) (1 arc-second SRTM DEM) but 0.36 in Perron and Royden (2013)

(1/3 arc-second USGS DEM). However, for averaging results (as done in Sect. 4.3), uncertainty in elevation data may not cause distinct differences in parameter estimates in this study area (e.g. $\theta$ and $k_{sn}$)." in Line 12-16 P10.

P. 9, line 19: Say "The knickpoint is located at an elevation of…"

5   Thanks for the comment. We have revised the text as "The knickpoint is located at an elevation of ~280 m" in Line 25 P10.

P.9, line 24: replace "it" before "suggests" with "these steepness indices suggest".

Thanks for the comment. We have revised the text as "these steepness indices suggest that other variables" in Line 30 P10.

10   P.9, line 32: I don't feel the sentences starting with "thus" is supported. Why would you trust the S_A analysis more than the chi analysis when you have already shown that estimates of k_sn from S-A analysis are highly uncertain. I believe your results show that when the S-A analysis differs from the chi analysis it should be the chi analysis that should be trusted due to its lower uncertainties. Either explain or delete this sentence.

Thanks for the comment. We have deleted it.

15   We want to show that a river may be in disequilibrium condition despite a linear relationship in the χ-z plot. We have added sentences in Line 9-14 P11. The text is as below:

Both the slope-area data (Fig. 17b) and the χ-z plot based on a $\theta$ value derived from collinearity test (Fig. 18c) detect the unsteady signal (a knickpoint located at an elevation of ~280 m) on the trunk stream of the Mattole river, despite the best linearity for the integral approach (Fig. 17e). Then, we can find that a river may be in disequilibrium condition in spite of a

20   linear relationship in the χ-z plot. In some cases, uplift can be inserted along rivers, which makes values of χ difficult to interpret (Czarnota et al.,2014;Paul et al., 2014;Pritchard et al., 2009; Roberts and White, 2010; Roberts et al., 2012; Wilson et al., 2014).

P. 10 line 22: Does "Amanda" have a surname?

25   Thanks for the comment. We have added her surname as "Amanda McDowell" in Line 30 P11.

[revised manuscript text omitted]

---

## Author Response (AR3)

**Associate Editor Decision: Publish subject to minor revisions (review by Editor) (27 Jan 2017) by Prof Simon Mudd**

Comments to the Author (pdf): esurf-2016-40-comments-to-author.pdf

Comments to the Author:

I have a few minor edits that should make the text clearer. Hopefully it takes no more than half an hour. My comments are

5   attached in an annotated pdf.

I look forward to seeing this paper in ESURF.

Thanks for the comments and we have revised it in the text directly. See details as below:

In Line 8-9 P1, we have revised the text as "produces stream power parameters with high uncertainties relative to the integral

10   approach".

In Line 15 P2, we have revised the text as "Differentiation leads to considerable scatter in slope-area plots".

In Line 28-31 P2, we have revised the text as "because unlike the integral method one does not have to set an $m/n$ ratio, but rather measures this ratio directly from topographic data. In addition, the uncertainty in $k_s$ will be underestimated using the integral method".

15   In Line 10 P3, we have revised the text as "is frequently debris flow dominated".

In Line 28 P3, we have revised the text as "Here, we combine the Durbin-Watson test".

In Line 7 P5, we have revised the text as "We divided the profile of Cooskie stream into colluvial and bedrock channels from the log-transformed slope-area plot by eye (Fig. 3a)".

In Line 9 P5, we have revised the text as "For example, Mudd et al (2014) used a segmentation algorithm".

20   In Line 3 P7, we have revised the text as "slope-area plots make no assumptions about $\theta$ and therefore are more sensitive than $\chi$ analysis for detecting spatially varying concavities".

In Line 5 P7, we have revised the text as "Combining these methods with statistical tests".

In Line 25 P7, we have revised the text as "However, concavity varies in streams consisting of both bedrock and alluvial channels".

25   In Line 9-10 P10, we have revised the text as "which mirrors the findings of both Snyder et al (2000) and Wobus et al (2006). We find no distinct difference in concavity and channel steepness indices when using different datasets and data handling methods."

In Line 11 P10, we have revised the text as "Utilising different datasets may cause some differences in parameter estimate for an individual catchment."

30   In Line 13 P10, we have revised the text as "However, for averaged results".

In Line 19 P10, we have revised the text as "To explore the effect of landscape transience, we analyzed Mattole River, a large river in the MTJ region (Fig. 17a)."

In Line 21-22 P10, we have revised the text as "Using a 300 m smoothing window and 20 m contour sampling interval, we derived a log-transformed slope-area plot of the stem (Fig. 17b). We recognized the critical threshold of drainage area, $A_{cr}$, ~0.1 km$^2$ and at the elevation of ~450 m (Figs. 17b and c) from the slope-area plot by eye."

In Line 32-33 P10, we have revised the text as "However, channel may be transient in which case previous authors have suggested either segmentation of $\chi$ profiles (Mudd et al., 2014) or interpretation through inversion methods (e.g., Goren et al., 2014).

In Table 1 P15, we have revised the text as "Positively auto-correlated" and "Negatively auto-correlated".

[revised manuscript text omitted]